# Global glacier-free topography reveals a large potential for future lakes in presently ice-covered terrain

T. Frank [1] ✉, W. J. J. van Pelt [1], D. R. Rounce [2], G. Jouvet [3] & R. Hock [4,5]

Glacier retreat transforms landscapes in polar and mountainous regions. Yet, the topography of the emerging terrain remains poorly known. Here, we present a physically consistent, global map of the ice-covered topography beneath all glaciers on Earth distinct from the ice sheets, derived from the three-dimensional higher-order Instructed Glacier Model, and constrained by extensive observational datasets. The map allows us to identify > 50,000 possible future lakes in the presently ice-covered landscape, with a maximum total volume of 3,138 km$^3$—enough to store 7 mm sea-level equivalent (SLE). Additionally, we estimate the total global glacier volume at $149.41 \pm 29.28 \times 10^3$ km$^3$ ($308 \pm 60$ mm SLE). Large overdeepenings near glacier fronts in High Mountain Asia suggest an increased risk for glacier lake outburst floods under glacier retreat. The subglacial topography and ice thickness data offer new opportunities for diverse cryospheric and Earth system studies, including refined projections of glacier changes and landscape evolution of deglaciated terrain.

Glaciers, distinct from the large ice sheets in Greenland and Antarctica, are key components of the Earth system, yet are critically threatened by anthropogenic climate warming[1,2]. The negative impacts on human societies and the natural world are numerous[3,4], including accelerating sea level rise—where glacier melt currently contributes more than each of the much larger Greenland and Antarctic ice sheets[2]—and the loss of critical water resources acting as a buffer against water stress across Asia and South America[5]. A rapid transition to a net-zero economy would substantially reduce these impacts, and ambitious climate policies in line with the 1.5 °C Paris agreement goal could double long-term glacier preservation compared to current policies[6,7]. Nonetheless, even under such an optimistic climate scenario, projections indicate a loss of 26% of glacier mass and 21% of glacier area by the end of the century[1]. These values increase to 43% and 42%, respectively, for the high-end SSP5-8.5 scenario, with almost complete deglaciation projected for several large-scale glacier regions[1].

Indeed, land has already started to emerge from the retreating ice unveiling >2000 km of previously ice-covered coastline in the Northern Hemisphere alone in the last 20 years[8]—a process that is expected to accelerate[1]. However, the topography of the emerging landscape is poorly known, which poses a severe limitation to our ability to sustainably manage these lands, e.g., for hydropower production[9], to characterize and, where appropriate, protect arising ecological niches[10], or to mitigate potential hazards, such as Glacier Lake Outburst Floods (GLOFs) from moraine- or bedrock-dammed lakes[11,12]. Insufficient knowledge of subglacial topography also hampers accurate projections of future glacier retreat because bed shape is a key input for ice flow models that predict future glacier change[7]. Moreover, total ice volume derived from bed and surface elevations is an important variable to inform projections of sea level rise and freshwater availability.

No previous study has directly reconstructed global subglacial topography, although Farinotti et al.[13] and Millan et al.[14] estimated glacier thicknesses, allowing bed elevations to be inferred by subtracting these values from surface topography. Despite showing close agreement in global glacier volume—$32 \pm 8$ and $31 \pm 10$ cm sea level

[1]Department of Earth Sciences, Uppsala University, Uppsala, Sweden. [2]Department of Civil and Environmental Engineering, Carnegie Mellon University, Pittsburgh, PA, USA. [3]Institute of Earth Surface Dynamics, University of Lausanne, Lausanne, Switzerland. [4]Department of Geosciences, University of Oslo, Oslo, Norway. [5]Geophysical Institute, University of Alaska Fairbanks, Fairbanks, AK, USA. ✉e-mail: thomas.frank@geo.uu.se

equivalent (SLE), respectively, after homogenization[15]—regional discrepancies exceeding 25% and often markedly different bed maps between the two studies highlight considerable uncertainties. These reflect specific methodological limitations, particularly the reliance on the shallow ice approximation (SIA). This formulation is valid only for small depth-to-width-ratio geometries typical of large glaciers and ice sheets with reduced basal motion[16], but has traditionally been applied in global-scale glacier studies due to computational constraints[17]. Additional simplifications, such as the use of 1D flow-line models[13] or difficulties in constraining ice velocities on slow-flowing glaciers[14], have also contributed to demonstrably unrealistic bed geometries (see below)[18,19].

Here, we overcome these limitations by applying a 3D higher-order ice flow model[20] at global scale, together with a rigorous data processing pipeline. Specifically, we use the Instructed Glacier Model (IGM) accelerated by GPU and deep learning[21] in a bed inversion framework that directly reconstructs distributed subglacial topography (Methods)[22]. We thus produce a globally consistent and physically realistic map of glacier beds, the Topography of a Deglaciated Earth (TOPO-DE v1.0). Leveraging more input datasets than previous studies, automatically calibrating against all available thickness observations, and explicitly modeling basal sliding, we reconcile physically consistent ice dynamics, observational constraints, and subglacial topography. Taking advantage of the improved bed product, we provide a global assessment of potential future lakes in a deglaciated world with important implications for sea level rise, glacier hazards and landscape evolution, and we refine estimates of global and regional glacier volumes.

## Results
### Unveiling the topography beneath the world's glaciers
Since direct observations of glacier bed elevations are only available for ~2% of glaciers globally[23], spatially complete subglacial topography must be inferred through inversion methods constrained by surface data. We have compiled a comprehensive dataset from publicly available sources for the > 200,000 glaciers in the globally complete Randolph Glacier Inventory v6.0 (RGI)[24,25], including digital elevation models (DEMs)[26], glacier outlines[24,25], surface elevation changes[27], ice flow velocities[14,28], and modeled specific mass balances[1], alongside frontal ablation estimates for marine-terminating glaciers in the Northern Hemisphere[29] and Patagonia[30]. This data feeds into an iterative inversion workflow that matches observed and modeled surface elevation changes, previously validated on synthetic glaciers[22] and successfully applied on regional scales[19,31]. In the algorithm, IGM runs for at least 2000 model years per glacier—an effort computationally prohibitive with traditional models of comparable complexity at global scale, now enabled by GPU acceleration (Methods)[21]. Glaciers that share boundaries are treated as a single entity to avoid artificial discontinuities in bed topography observed in previous products[13]. In fast-flowing sectors of marine-terminating glaciers, a spatially distributed friction parameter is inverted to ensure consistency between modeled and observed ice flow velocities and prevent unrealistic thicknesses[22]. Automatic Bayesian calibration of model parameters[32] against all > 3,800,000 point observations of ice thickness in the Glacier Thickness Database v4-beta (GlaThiDa)[23,33] for each of the 19 RGI glacier regions with available data, ensures that region-wide parameters yield the best inversion performance.

### Bed topographies for all glaciers on Earth
The details of our new global subglacial topography dataset cannot be adequately visualized here, so we refer to the openly available dataset at https://doi.org/10.6084/m9.figshare.29940932[34] and showcase here only example outputs of a valley glacier and an ice cap (Fig. 1). The modeled topographies (100−400 m resolution) align with our expectations of subglacial morphology, revealing characteristic features such as U-shaped valleys, glacial cirques, and overdeepenings. Similarly, well-known features of marine-terminating glaciers, such as retrograde bed slopes and deep basins below sea level, are well represented, for example, in the Arctic, Antarctic, and Patagonia. These glaciers currently comprise ~ 60% of the global ice volume (see below) despite accounting for < 2% of the glacier count. Given that their retreat is highly sensitive to bed geometry[35,36], our new topographies are expected to substantially improve future sea-level projections.

The strength of TOPO-DE lies in its physical consistency, achieved through a higher-order representation of ice flow dynamics combined with an inversion method based on mass conservation. Error metrics against observed thicknesses indicate improved skill compared to previous studies (Fig. S1, S2), although these metrics should be interpreted with caution because a shared, independent validation dataset across studies is lacking (Supplementary Discussion). Importantly, TOPO-DE substantially reduces systematic biases found in earlier thickness estimates which tended to overestimate thin and underestimate thick ice (Fig. 2). This has important implications for accurate glacier projections: With TOPO-DE, thinner glaciers with smaller initial thickness will disappear faster under future warming compared to using previous (greater) thickness estimates. Thicker glaciers, by contrast, are expected to survive longer. These biases would naturally also affect the mapping of potential future lakes. For example, unrealistically smooth beds may be caused by underestimating thickness variability within individual glaciers, which could imply that the potential for lakes is underestimated.

TOPO-DE also visibly improves the realism of glacier beds (Figs. S3, S4). Previous reconstructions frequently produced features that are physically implausible—such as "walls" between individual outlet glaciers of ice caps, highly irregular beds in slow-flowing areas, "stripes" perpendicular to the flow direction or flat beds under ice caps (Supplementary Discussion). Such erroneously mapped features can have significant practical implications. For instance, they can prevent the correct delineation of emerging watersheds as glaciers retreat, and/or induce numerical stability issues in future glacier simulations. In contrast, our results are consistent with our physical understanding of subglacial morphology (Figs. S3, S4). Moreover, when paired with geological maps of surrounding unglaciated terrain, our glacier bed maps provide valuable insights into subglacial geology. For instance, a sequence of resistant and weak rocks mapped outside a glacier in the Canadian Arctic[37] can now be traced in detail beneath the ice, whereas the same formations were weakly expressed or absent in previous products (Fig. 5). This highlights the potential of TOPO-DE to aid geological mapping of currently ice-covered landscapes.

Nevertheless, TOPO-DE inevitably contains inaccuracies arising from input data errors, physical shortcomings, and solution equifinality inherent to underconstrained ice thickness inversions (Supplementary Discussion). Biases in surface elevation change, mass balance, glacier outlines, velocities, and DEMs can locally affect inferred thickness and bed geometry[22,38]. Meanwhile, the use of higher-order ice-flow physics substantially reduces physical shortcomings compared to SIA-based approaches[13,14]. Errors are expected to be elevated for surging glaciers[39], in regions with few thickness and mass balance observations[40,41], and for bed features smaller than approximately one ice thickness, which are fundamentally unconstrainable[42-44]. More ice thickness observations are critical to improve glacier bed inversions further, specifically in data-sparse regions such as High Mountain Asia, the Russian Arctic, and Subantarctic & Antarctic Islands. Our input-preparation and data-assimilation framework effectively minimizes and balances error sources, but residual artifacts naturally remain and can propagate into modeled lake locations and volumes. At large scales we expect errors to average out, but localized applications should evaluate bed shape, ice thickness and potential lake locations in light of these uncertainties. Importantly, the data assimilation

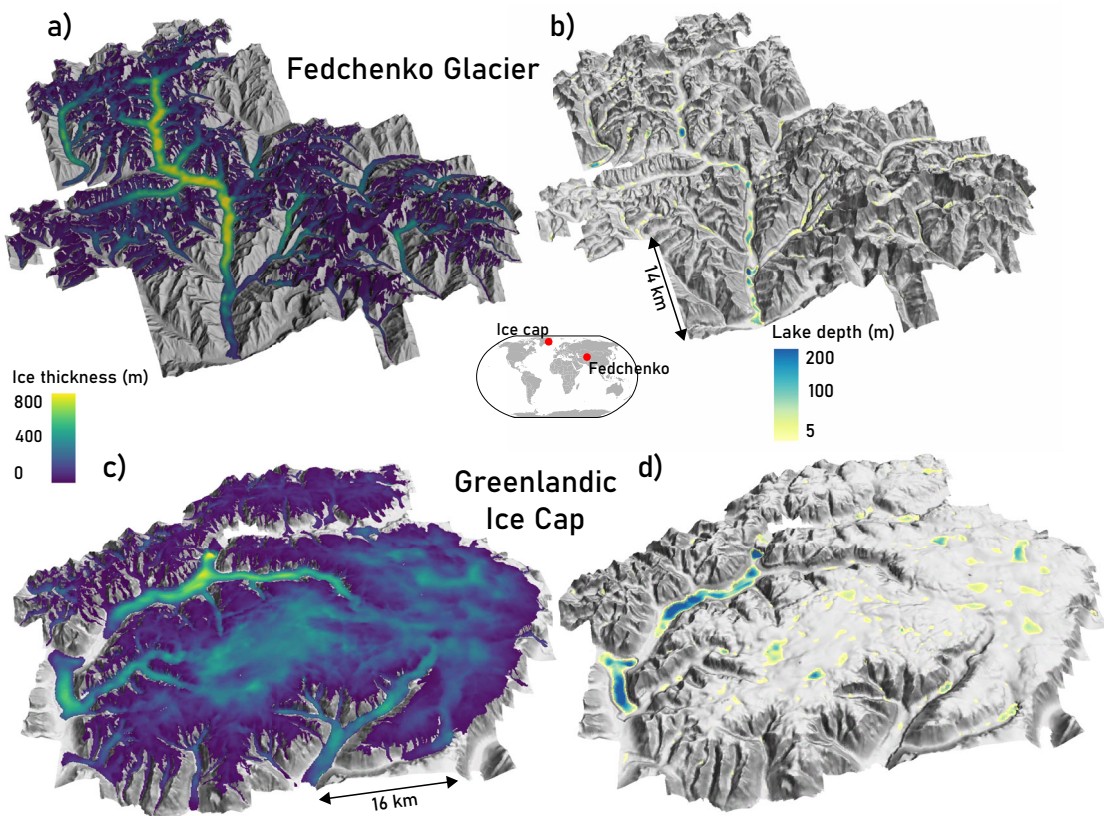

**Fig. 1 | Examples of model output for Fedchenko glacier in Central Asia (central coordinates: 72.23°E 38.80°N) and an ice cap in Tuttut Nunaat, Greenland (26.78°W 71.22°N). a**, **c** Modeled ice thickness draped over present-day surface topography; **b**, **d** glacier-free topography with future lakes. Topographies in all plots are shown as hillshades.

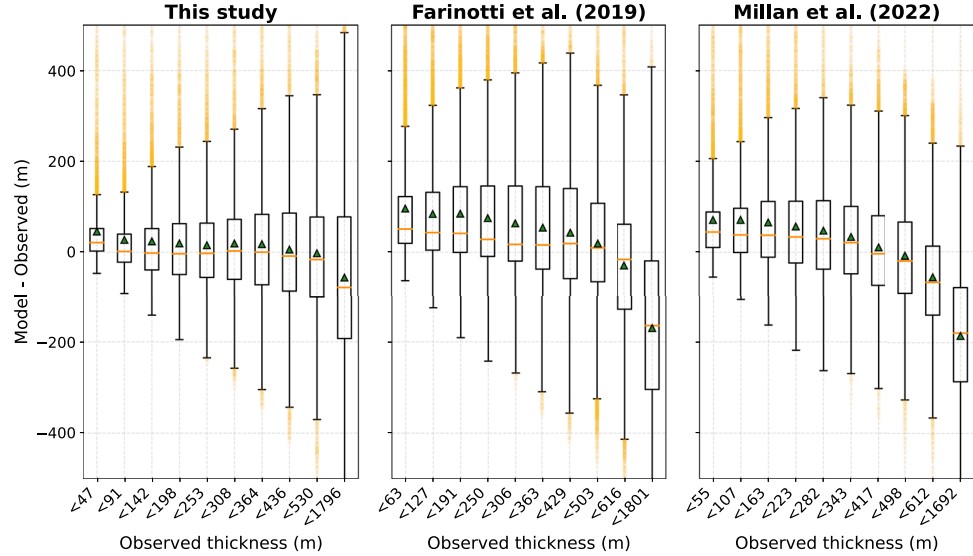

**Fig. 2 | Boxplot of the model-observation misfit for ten thickness classes, each containing 10% of the data, for this product and the thickness products by refs. 13, 14.** Horizontal orange lines represent the median and triangles the mean misfit, the box represents the interquartile range (Q1–3), the whiskers 1.5 times the interquartile range and orange dots represent fliers. Note that the thickness classes differ slightly due to the different spatial coverage of the products. The same analysis with uniform classes yields the same patterns (not shown).

framework used here makes TOPO-DE particularly well-suited to be refined by future improvements in input datasets.

## Detecting potential future lakes

Leveraging the physically realistic bed shapes, we predict the locations and volumes of new lakes that would form if all glacier ice was to melt.

This is done by mapping overdeepenings in the modeled subglacial topographies and calculating their volume up to the height at which water would overflow to adjacent grid cells. For simplicity, we assume no modification of bed shape by future geomorphic processes, a simplification which likely renders our areas and volumes upper-bound due to dam breaches and sediment infill that reduce the initial

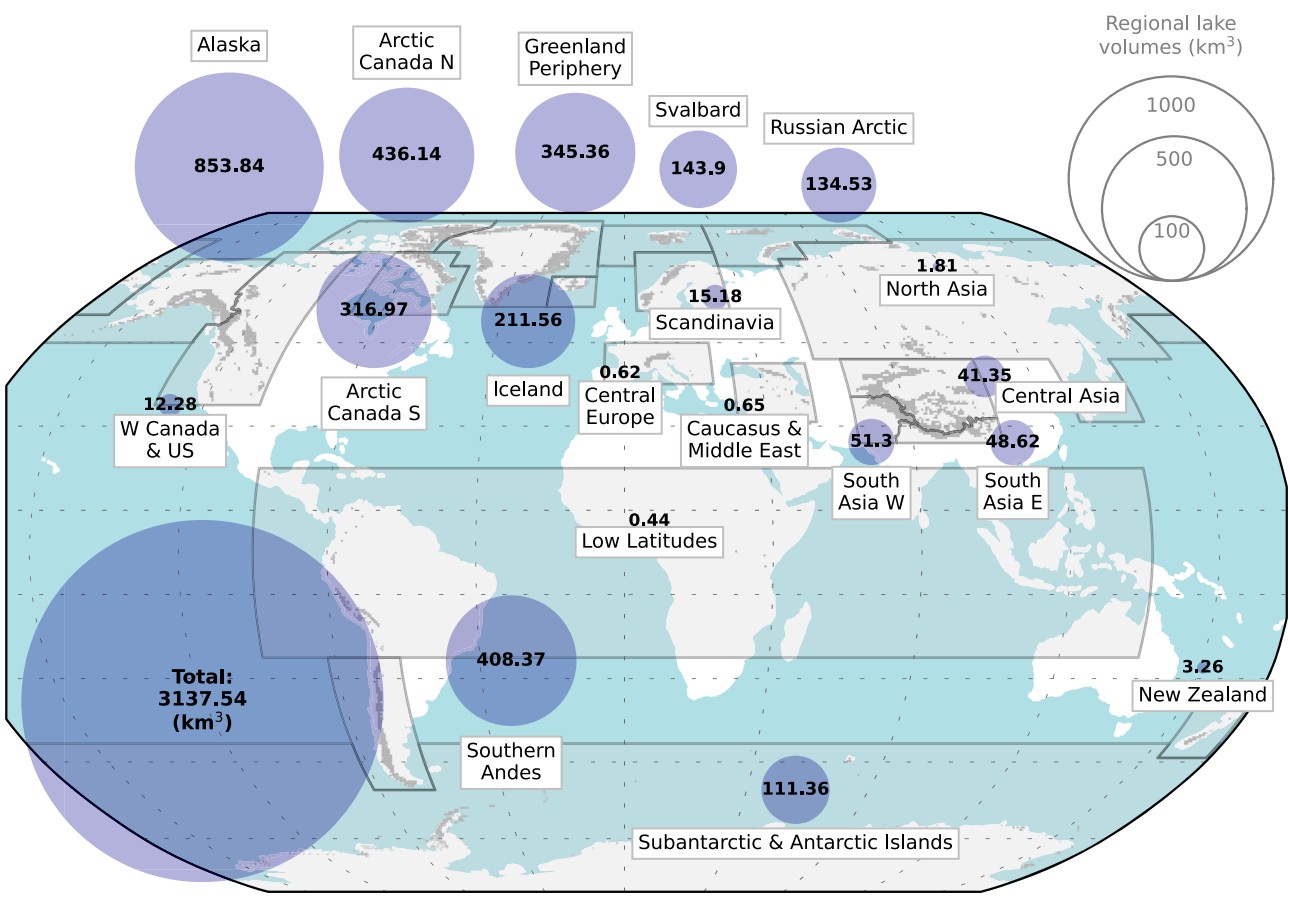

**Fig. 3 | Global and regional potential lake volumes in km³ under present-day ice cover by Randolph Glacier Inventory glacier region.** Shaded polygons show Randolph Glacier Inventory glacier regions. Glacier cover is indicated in light grey.

lake potential over time[12]. In addition, unresolved small-scale topography may include drainage pathways not represented here.

Globally, we estimate a potential for 56,659 new lakes (≥0.05 km²) with an area of 40,647 km² (Fig. 3, Table 1). Lakes would cover ~ 6% of the newly deglaciated landscape—approximately three times the proportion of lake cover on global present-day non-glacierized land (2%[45]). In the Alps, refs. 46,47 observed lake covers of 0.5% (Austria) and 0.9% (Switzerland) on land that became ice-free since the Little Ice Age. We estimate a potential future lake coverage of 0.9% in Central Europe, closely aligned with these historic values. For comparison, ref. 48 modeled a potential future lake area of 45.2 km² in the Swiss Alps, corresponding to a coverage of 4.7%, about four times higher than the historical reference. In Svalbard, ref. 49 showed an increase in ice-marginal lakes of 72 km² between the 1930s and ~2010 while 2144 km² land became ice-free[50] (3.3% lake coverage). Here, our estimate is 5.6% (including not only ice-marginal, but also proglacial lakes). The lakes would globally be capable of holding 3138 km³ of water (Fig. 3, Table 1). Considering only the volume above sea level, the lakes would lower the potential sea-level contribution of complete global glacier melt by 7 mm (2%). This amount is double the sea-level equivalent stored in present-day glacial lakes (defined as lakes hydrologically connected to a glacier and within 10 km of the RGI outline)[51]. The global mean depth of the potential lakes is 77 m, the mean lake surface elevation is 711 m above sea level, and the average overlying ice thickness is 410 m—approximately twice the mean global ice thickness (see below).

The largest potential lake volumes are found in Alaska (854 km³), the Southern Andes (408 km³), and Arctic Canada North (436 km³), regions that are already known to host many glacial lakes today[12]. The first two regions also show the largest mean lake depths (117 and 124 m,

respectively). In this regard, Arctic Canada North (mean depth: 60 m) is more similar to other high-latitude regions such as Greenland, Svalbard, Subantarctic & Antarctic Islands and Arctic Canada South, where the mean depths range between 60 and 75 m (Table 1). The deep lakes found in Alaska and the Southern Andes, but also in Iceland (mean depth: 97 m) and New Zealand (mean depth: 96 m), coincide with maritime conditions in these regions. Maritime temperate glaciers have a large erosional potential, thus efficiently creating bedrock overdeepenings and tall moraines capable of damming lakes[52]. New Zealand has potential for ~ 5.5 times more new lake volume than the more continental region of Central Europe, despite the latter having roughly twice the ice volume and glacierized area (Fig. 3). The largest mean lake areas are found in Iceland (2.73 km²), the Russian Arctic (1.17 km²), and Alaska (1.12 km²) whereas North Asia and Low Latitudes —both home to small mountain glaciers—exhibit the on average smallest lakes (0.10 km²). In terms of lake density per glacierized area, we establish a link to current glacier geometries. Excluding tidewater glaciers, we find that regions with steeper glacier surfaces have smaller potential lake volumes per glacierized area than regions with gentler slopes ($r = -0.65$, $p < 0.05$; Fig. S6). High-latitude regions with often thick glaciers extending to sea-level show smaller surface slopes and correspondingly larger potential lake volumes per area. Conversely, the glacierized mountains of the mid-latitudes in North America, Europe and Asia are characterized by steep surface topography and comparatively small lake volumes per area. Despite this relative sparsity of lakes in High Mountain Asia, the mean lake depth there (51 to 79 m) is comparable to flatter polar regions and considerably greater than in other steep mountain regions (Table 1). A likely explanation are tall moraines due to the extensive debris cover of many glaciers[53] that allow deep lakes to form.

**Table 1 | Statistics of potential future lakes in the presently glacierized area if glacier ice were removed**

| Region | Area (km²) | n | Volume (km³) | Mean depth (m) | Mean area (km²) | Mean lake surface elevation (m) | Mean overlying ice thickness (m) |
|---|---|---|---|---|---|---|---|
| 01-Alaska | 7289.2 | 6385 | 853.8 | 117.1 | 1.12 | 640 | 546 |
| 02-W Canada & US | 314.4 | 1605 | 12.3 | 39.1 | 0.20 | 1578 | 258 |
| 03-Arctic Canada N | 7246.9 | 8859 | 436.1 | 60.2 | 0.69 | 462 | 352 |
| 04-Arctic Canada S | 4721.1 | 5409 | 317.0 | 67.1 | 0.86 | 452 | 342 |
| 05-Greenland Periphery | 5936.9 | 11518 | 345.4 | 58.2 | 0.52 | 518 | 332 |
| 06-Iceland | 2182.5 | 797 | 211.6 | 96.9 | 2.73 | 618 | 452 |
| 07-Svalbard | 1907.2 | 1491 | 143.9 | 75.5 | 1.10 | 105 | 364 |
| 08-Scandinavia | 339.8 | 1333 | 15.2 | 44.7 | 0.22 | 1129 | 229 |
| 09-Russian Arctic | 3219.6 | 2756 | 134.5 | 41.8 | 1.17 | 210 | 308 |
| 10-North Asia | 51.2 | 521 | 1.8 | 35.4 | 0.10 | 1577 | 144 |
| 11-Central Europe | 19.7 | 143 | 0.6 | 31.5 | 0.14 | 2337 | 216 |
| 12-Caucasus & Middle East | 15.4 | 84 | 0.7 | 42.4 | 0.15 | 3034 | 192 |
| 13-Central Asia | 817.0 | 4602 | 41.3 | 50.6 | 0.18 | 4477 | 231 |
| 14-South Asia W | 799.5 | 2341 | 51.3 | 64.2 | 0.20 | 4433 | 319 |
| 15-South Asia E | 616.4 | 2466 | 48.6 | 78.9 | 0.25 | 4755 | 281 |
| 16-Low Latitudes | 14.5 | 144 | 0.4 | 30.5 | 0.10 | 4942 | 100 |
| 17-Southern Andes | 3288.2 | 3246 | 408.4 | 124.2 | 1.01 | 559 | 691 |
| 18-New Zealand | 34.1 | 89 | 3.3 | 95.7 | 0.38 | 916 | 238 |
| 19-Subantarctic & Antarctic Islands | 1833.1 | 2870 | 111.4 | 60.7 | 0.59 | 185 | 432 |
| Global | 40646.7 | 56659 | 3137.6 | 77.2 | 0.91 | 711 | 410 |

Global and regional potential future total lake area, number (*n*), volume, mean depth, mean area, mean of the lake surface elevation (i.e. the elevation at which the lake surface would be if the ice were removed) and the mean overlying ice thickness above the lake bottom.

We find the largest potential lake volumes in High Mountain Asia in the lowermost parts of the glaciers (Fig. 4). Although our bed maps do not indicate dam types and materials, this suggests that a large proportion of potential lakes in High Mountain Asia is moraine-dammed, as are many present-day proglacial lakes in the region[54]. By number of individual lakes, we do not see a similar concentration at low elevations, indicating that the low-elevation lakes have above-average volumes. This makes them susceptible to generating large and potentially devastating GLOFs[12]. Similar lake distributions as in High Mountain Asia are found in Alaska and New Zealand (Fig. 4). Both regions are known to feature large proglacial lakes today[55], such as the proglacial lake of Tasman glacier, which we estimate has the potential to expand further by 17 km² [from 7 km² in 2014[55]]. Other regions dominated by valley glaciers (such as Central Europe, Caucasus & Middle East, and Low Latitudes) also show a concentration of large lakes relatively close to the glacier fronts, albeit to a somewhat lesser degree. This pattern likely results from the joint presence of lakes in bedrock and sediment along low-sloping glacier tongues in major valleys, and of moraine-dammed lakes. Meanwhile, polar regions such as Arctic Canada N, Greenland, Russian Arctic and Subantarctic & Antarctic Islands show more uniform lake distributions over the glacier altitudinal range. This is because beds of tidewater glaciers below sea level will not form lakes, and also likely due to the overall flatter topography, which is more prone to overdeepening anywhere along the glacier than in steep mountains (Fig. S6). In Scandinavia, Greenland and the Russian Arctic, lake volumes are comparatively more pronounced at higher relative elevations. Under hypothetical globally uniform relative glacier retreat rates, this implies that it would take the longest in these regions until the regional majority of potential lake volumes is realized. In the upper third of the glacier altitudinal range in most regions, we find numerous lakes though smaller volumes, likely reflecting the widespread presence of small lakes in glacial cirques (Fig. 4).

Historically, roughly one third of GLOFs were caused by drainage of moraine- or bedrock-dammed lakes that we can identify in TOPO-

DE, with the other two thirds resulting from ice-dam failure[12]. In High Mountain Asia, catastrophic GLOF events from moraine-dammed lakes have occurred frequently and human exposure to GLOF impacts is the highest globally[12,56,57]. Although so far reporting biases preclude the detection of significant positive GLOF trends related to glacier retreat[12], the future presence of large lakes near the glacier margins strongly suggests that GLOF hazard from moraine-dammed lakes will increase[58]. Moreover, future hazards will likely also increase from emerging bedrock-dammed lakes which are less vulnerable to dam failure, and thus historically have caused fewer GLOFs, yet may release large volumes of water due to displacement waves from mass movements into the lakes[12]. Recent studies indicate increasingly unstable mountain flanks due to permafrost thaw and consequently more frequent mass movements, underscoring and exacerbating the substantial hazard emanating from all emerging lakes[59].

In High Mountain Asia, we find a total potential lake volume of 141 km³ (Table 1), roughly consistent with 120 km³ estimated by ref. 60. However, our estimate is around 2.5 and three times higher than estimates by ref. 61 (60 km³) and ref. 62 (50 km³), respectively, with both previous analyses relying on the bed topographies of ref. 13. References 61, 62 applied morphological criteria such as surface slope thresholds in addition to a sink-fill algorithm to identify potential future lakes, naturally rendering their estimates more conservative than ours[47]. However, the known artifacts in ref. 13 dataset and its possible tendency to exhibit overly smooth beds (see above and ref. 19) may have led to an underestimation of potential lake volumes, and thus GLOF hazard. This highlights the importance of utilizing realistic bed shapes for lake mapping and hazard assessments. A priority for future studies on GLOF hazards (e.g.[63]) should be to synthesize and validate different available lake products on the local scale, thus increasing confidence in modeled lake locations and volumes, and ultimately enabling effective mitigation measures[57].

Beyond hazards, emerging lakes may support water storage, tourism[64], and hydropower generation[9]. For instance, a recently emerged proglacial lake in a glacier-carved overdeepening at Trift

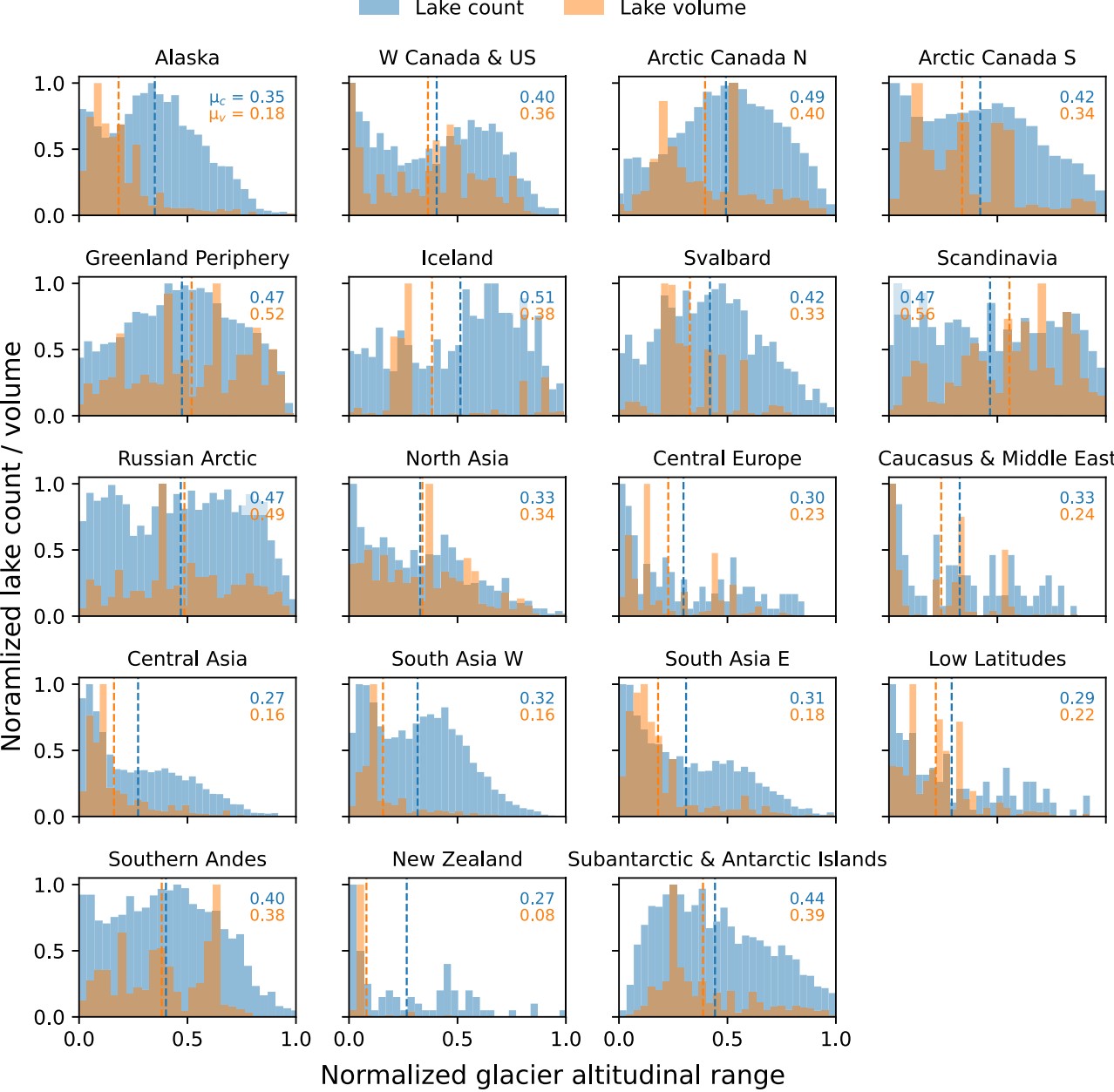

**Fig. 4 | Lake distribution over glacier altitudinal range for each glacier region.** Number of lakes (blue) and lake volume (orange) in 30 elevation bins, with the $y$ axis normalized to the largest bin. The normalized glacier altitudinal range represents a scale from 0 (lowest) to 1 (the highest point of a glacier). Each lake's location on that scale is computed by normalizing the mean glacier surface elevation under which the lake is located. Blue and orange numbers in each subplot show the mean normalized altitude of the lake counts ($\mu_C$) and volumes ($\mu_V$), respectively.

glacier, Switzerland, has become a popular tourist attraction and is a suggested site for hydropower production[65]. For the sustainable development of each site, case-specific opportunities and constraints must be carefully assessed, taking into account various factors including technological feasibility, environmental vulnerability, and social acceptance[58,65].

**Global and regional ice volumes**

Knowledge of subglacial topography combined with surface DEMs directly yields ice volumes, here aggregated regionally, globally, and by latitude for approximately the year 2013 (Fig. 5, Table S1). We find a global glacier volume $V_{total} = 149.41 \pm 29.28 \times 10^3 \, km^3$, corresponding to a SLE of $308 \pm 60 \, mm$ accounting only for ice volume above flotation, and a mean thickness of $212 \pm 41 \, m$. This estimate agrees well with the two most recent independent studies[13,14] and consequently further

constrains global glacier volume (Fig. 6). Earlier estimates were higher by up to a factor of two due to considerably simpler methodological approaches and a lack of high-quality input data, and now appear increasingly unlikely[15,18,66–70].

Regionally, our results show marked differences from the most recent previous studies[13,14]. In the largest RGI region, Subantarctic & Antarctic Islands, our volume of $50.57 \pm 8.34 \times 10^3 \, km^3$ closely matches the $46.47 \pm 12.06 \times 10^3 \, km^3$ reported by ref. 13, but exceeds that of ref. 14 ($35.10 \pm 9.10 \times 10^3 \, km^3$) by $> 40\%$. The considerable disagreement with ref. 14, where uncertainties barely overlap, may be a result of ref. 14 predominantly relying on BedMachine[71] in this region, which likely produces too thin ice in poorly observed valley glaciers, as previously shown for the Antarctic peninsula[72]. In contrast, for the Greenland Periphery, our volume ($12.46 \pm 0.94 \times 10^3 \, km^3$) is nearly identical to that of ref. 14 ($12.54 \pm 3.95 \times 10^3 \, km^3$), but ~20% lower than

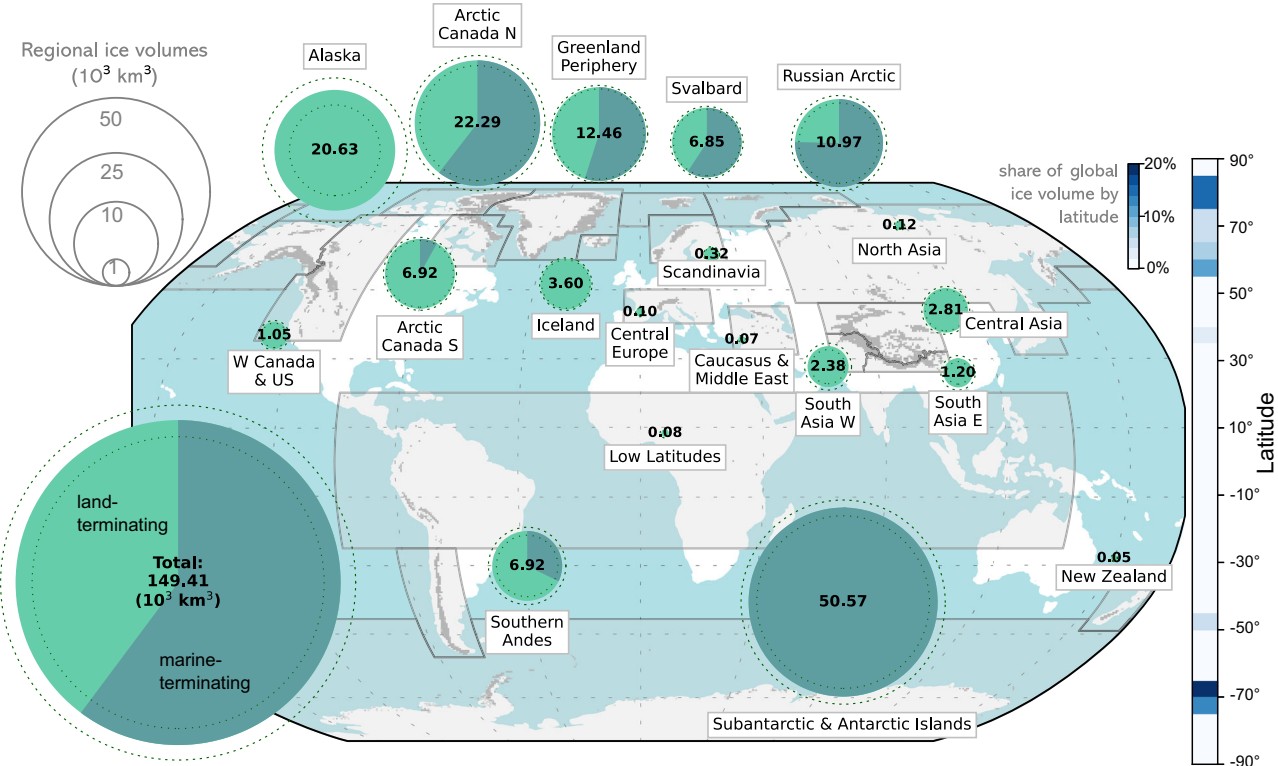

**Fig. 5 | Global and regional glacier volumes in 10³ km³.** Dashed circles indicate volume uncertainties, and dark shaded pies the volume stored in marine-terminating glaciers (as defined in the Randolph Glacier Inventory; in Subantarctic & Antarctic Islands, also including shelf-terminating). Shaded polygons show Randolph Glacier Inventory glacier regions. Glacier cover is indicated in light grey. The blue shaded bar to the right indicates the share of global glacier volume by latitude (5° bins).

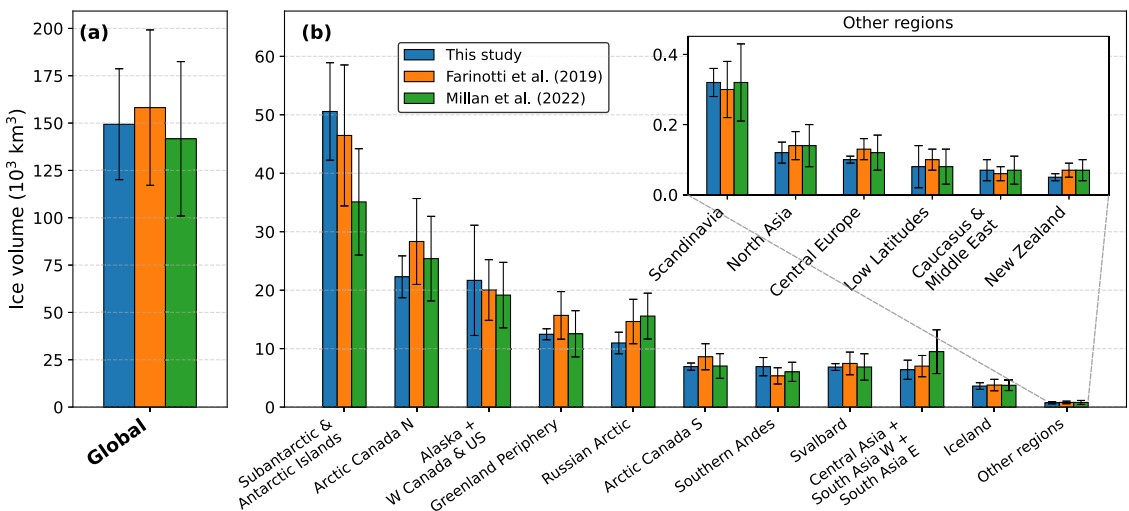

**Fig. 6 | Global and regional ice volume comparison. a** Global and **b** regional ice volume from this study, refs. 13 and 14 including uncertainty bars. Regions with the smallest volumes are shown separately. Note that the volumes by ref. 14 refer to the homogenized and corrected numbers calculated by ref. 15.

the $15.69 \pm 4.07 \times 10^3$ km³ estimated by ref. 13, with their central estimate outside our uncertainty range. Reference 13 is biased by ~90 m to thickness observations in this region, whereas our results are almost bias-free (4 m, Fig. S1). These examples from two regions highlight how our results provide much-needed clarification where the two previous estimates diverged. Overall, our volumes agree more closely with ref. 14 than with ref. 13 in the majority of regions. In High Mountain Asia, where glaciers act as critical water buffers[5], ref. 14 reported a volume 37% higher than ref. 13, suggesting that this would delay peak water by several decades. In contrast, we find a volume of

$6.39 \pm 1.64 \times 10^3$ km³, even ~9% lower than ref. 13. However, this region remains highly uncertain, with thickness observations available for only 16 out of >70,000 glaciers[23,41]. Future field campaigns are therefore critical to obtain more thickness observations for calibration and uncertainty reduction. In Arctic Canada North and the Russian Arctic, our volumes are between 12% and up to 30% lower than previous estimates, despite being negatively biased or unbiased relative to thickness observations (Fig. S1). This may directly reflect the difference between a higher-order and an SIA model, the latter requiring manual corrections in flat areas of ice caps (Fig. S3). However, the presence of

numerous surging glaciers in the Northern Canadian Arctic introduces uncertainties in both approaches.

## Implications and future perspectives

The new subglacial topographies provide critical data for a wide range of disciplines, including glaciology, hydrology, ecology, geology, and geomorphology. As glaciers retreat in a warming climate, improved knowledge of emerging landscapes and lakes is essential for land management, hydropower planning, safety in glacier tourism, as well as hazard and ecosystem change assessments. TOPO-DE offers substantial opportunities for further characterization of the subglacial landscape, which warrant dedicated investigation in future studies. Our large-scale analysis highlights major GLOF hazards associated with potential future lakes in steep mountain terrain, particularly large moraine-dammed lakes in High Mountain Asia. More detailed, localized investigations will be essential to further refine this picture and to generate actionable data for mitigating such mountain hazards under climate change. Future glacier simulations based on the new bed product will be instrumental in constraining the timing of glacier retreat and the associated emergence of proglacial lakes. Together with the revised ice-volume estimates, which diverge substantially from previous studies across many glacier regions, future glacier projections based on TOPO-DE are expected to reveal new regional patterns of deglaciation with associated improved estimates of sea level rise.

More broadly, our results underscore the value of computationally efficient, higher-order ice-flow models in inversion frameworks that exploit the growing availability of high-resolution remote sensing data, paving the way for global glacier modeling beyond the era of shallow ice flow physics and one-dimensional flow-line models.

## Methods

### Inversion methodology

The inversion methodology follows refs. [19,22,31], and operates by iteratively aligning modeled (mod) and observed (obs) rate of surface elevation change $dh/dt$. At each iteration $i$, the bed topography $B$ is updated as:

$$B^{i+1} = B^i - \left( \frac{dh^i_{\mathrm{mod}}}{dt} - \frac{dh_{\mathrm{obs}}}{dt} \right). \tag{1}$$

Whereas $dh/dt_{obs}$ is from ref. [27], $dh/dt_{mod}$ is obtained from running the distributed ice-flow model IGM v2.2.1 in an isothermal configuration[21]. The model domain is defined by glacier outlines from the RGI v6.0[24,25], and model forcing comes from the climatic mass balance product by ref. [1]. Input surface geometry is from the 90 m Copernicus DEM[26]. Our approach is mathematically identical to updating bed topography so that the flux divergence approximates the apparent mass balance—that is, the climatic mass balance minus $dh/dt$—once the algorithm has converged[19]. IGM solves higher-order ice flow equations[20] using physics-informed deep learning, delivering accurate solutions up to several orders of magnitude faster than traditional models, thanks to GPU acceleration[21]. To achieve convergence, the duration of the simulation should exceed a given glacier's response time[22]. We therefore model at least 2000, but up to 5000 yrs per glacier, with the number of iterations $i$, and hence bed updates, being determined by numerical stability criteria for solving the mass conservation equation[73]. The computational costs for this study correspond approximately to three weeks on six NVIDIA A40 GPUs. Given that traditional ice flow models of the same complexity are up to three orders of magnitude slower than IGM[73], this highlights the enormous computational advantages of GPU-based models here leveraged at global scale.

To regularize the inversion, the surface topography of the glacier is allowed to evolve such that a small fraction $\theta$ of each bed correction

is applied, albeit with opposite sign, at the surface:

$$S^{i+1} = S^i + \theta \left( \frac{dh^i_{\mathrm{mod}}}{dt} - \frac{dh_{\mathrm{obs}}}{dt} \right). \tag{2}$$

This formulation enables the model to absorb inconsistencies—such as input data errors or unmodeled processes—through modest adjustments to surface elevation rather than large, potentially unrealistic changes in bed topography.

In fast-flowing areas of marine-terminating glaciers basal friction is highly variable. Applying a spatially uniform friction coefficient would lead to erroneous bed reconstructions. Therefore, on such glaciers in areas where velocities exceed 100 m yr$^{-1}$, we additionally invert for a basal friction coefficient $c$ of a Weertman-type friction law implemented in IGM[21]:

$$c^{i+1} = c^i \times \left( 1 + F \left( \frac{u^i_{\mathrm{mod}} - u_{\mathrm{obs}} \cdot f_u}{u_{\mathrm{obs}} \cdot f_u} \right) \right), \tag{3}$$

where $F(x) = \min(\max(x, -\lambda), \lambda), \lambda = 0.8$ and $u$ denotes surface speed. This algorithm closely aligns modeled and observed surface speeds, while the scaling factor $f_u$ compensates for discrepancies where $u_{obs}$ cannot be matched with $u_{mod}$ without introducing thickness biases—for instance, due to $u_{obs}$ being seasonally biased. Velocity observations are sourced from ITS_LIVE[28] and ref. [14]. Friction inversions are performed once every 1000 model iterations, allowing the bed to adjust to a new friction field in between[22].

The parameters $\theta$, $f_u$, ice viscosity $\eta$ and a spatially uniform friction coefficient $c$ used in slow-flowing areas and on land-terminating glaciers are separately calibrated for each RGI region. In the Subantarctic & Antarctic Islands region, we also calibrate a region-wide frontal ablation rate $a$ as in ref. [13], due to a lack of observations. Calibration is performed using Bayesian optimization with Gaussian processes[32] to find the combination of parameters that minimize the target function:

$$J = \lambda_1 \frac{\mathrm{MAE}}{\bar{h}} + \lambda_2 \left| \frac{\mathrm{ME}}{\bar{h}} \right| + \lambda_3 \left| \frac{\overline{u_{mod}}}{u_{obs} \cdot f_u} \right|, \tag{4}$$

where $\bar{h}$ is regional mean ice thickness, and MAE and ME are the mean absolute and mean errors (bias) compared to GlaThiDa point thickness observations[23,33]. The weights are set to $\lambda_1 = 1, \lambda_2 = 1.5, \lambda_3 = 0.5$ in regions with marine-terminating glaciers and $\lambda_3 = 0$ elsewhere. This function balances precision (MAE), accuracy (ME), and dynamic consistency (modeled vs. observed velocity), while minimizing discontinuities across regions where $c$ is inferred dynamically (Eq. (3)) versus assigned uniformly. For regions lacking point bed observations parameter values are assigned based on neighboring regions. These regions are Iceland, where the mean of the parameters from Greenland and Svalbard are used, the Russian Arctic, where the Svalbard parameters are taken, and South Asia West, where the values from Central Asia are used. In Alaska, we apply manual tuning as automatic calibration leads to unrealistic ice thicknesses across most glaciers; this contributes to higher volume uncertainties there. For Scandinavia, we use the results of the methodologically nearly identical regional study by ref. [19]. Plots of observed vs. modeled thicknesses for each RGI region after calibration are shown in Fig. S1. Calibrated parameters per region are shown in Table S3. Given the time stamps of the input data, particularly the thickness observations, our calculated volumes refer approximately to the year 2013.

Simulations are performed at the level of glacier complexes—groups of immediately adjacent glaciers—to allow for mass exchange between connected glaciers. Grid resolution is glacier complex size dependent: small (area $A < = 7000$ km$^2$), medium ($7000$ km$^2 < A < = 28,000$ km$^2$) and large ($A > 28,000$ km$^2$) complexes

are modeled with 100 m, 200 m, and 400 m resolution, respectively. Coarser resolution for larger glacier complexes follows from the notion of larger glaciers generally being thicker[74], and thicker ice presenting a physical limit to the detail in bed topography that can be obtained with inversions, independent of chosen methodology[44].

Small ice-free areas may remain within the glacier outlines after the inversion due to mismatches between modeled and actual ice flow directions or inaccuracies in glacier outlines—both of which can allow ice to escape through the boundaries of the glacier domain. For land-terminating glaciers, thicknesses in such areas are reconstructed using a perfect-plasticity approach[75] with the yield stress derived from the ice-covered portions of the domain. For marine-terminating glaciers, where low surface slopes near the fronts would lead to unrealistically thick ice using perfect plasticity, these gaps are instead filled via bilinear interpolation. Glacial isostatic adjustment as a time-dependent process unfolding during and after deglaciation is not accounted for—the reconstructed beds reflect the present-day shape of the subglacial topography.

Technical details and schematic illustrations of the setup and input data preparation are given in the supplement.

## Uncertainty quantification

To estimate regional ice volume uncertainties, we note that $V = A\bar{h}$ where $A$ is the glacier-covered area and $V$ glacier volume. The corresponding volume uncertainty $\sigma_V$ is then given by:

$$\sigma_V = V \sqrt{\left(\frac{\sigma_A}{A}\right)^2 + \left(\frac{\sigma_{\bar{h}}}{\bar{h}}\right)^2}. \tag{5}$$

Area uncertainty $\sigma_A$ may vary across RGI regions, but conservatively, we here set a uniform relative uncertainty $\frac{\sigma_A}{A} = 5\%$, based on 1–2% and 3% estimated for Svalbard[76] and Norway[77], respectively.

Since all modeled regional mean thicknesses $\bar{h}$ are calibrated against thickness observations, their uncertainty $\sigma_{\bar{h}}$ arises entirely from the calibration process. We follow the approach of ref. 31 with additional terms to account for potential systematic biases in the thickness observations and in the representativeness of observed glaciers. Specifically, we include the following components:

1. Sampling uncertainty $\left(\frac{\sigma_C}{\sqrt{n_{obs}}}\right)$: The standard deviation of thickness biases $\sigma_C$ across glaciers with observations, divided by the square root of the number of observed glaciers $n_{obs}$, quantifies the uncertainty from calibrating an entire region based on a limited sample of glaciers. Conceptually, this is equivalent to the standard error of the mean. Based on four regions with many observed glaciers, we can show that $\sigma_C$ scales with $\bar{h}$ such that $\sigma_C = 0.3\bar{h}$ can be applied generally.
2. Calibration bias $\mu_C$: The residual bias between modeled and observed mean thicknesses across observed glaciers. By design of the calibration function $J$ (eq. (4)), $|\mu_C|$ should be small; however, a nonzero $|\mu_C|$ reflects a trade-off between competing calibration goals or a failure to identify the true minimum of eq. (4).
3. Systematic observation error $\sigma_{obs}$: A uniform 2 m error is added to account for potential systematic biases in the thickness observations themselves.
4. Representativeness error $\sigma_{repr}$: This term captures the error introduced by assuming that glaciers with observations are representative of all glaciers in the region—the key assumption of 1). This is important to consider as there may be geographical biases (e.g., all observed glaciers are concentrated in one area of a region) or glaciological biases (e.g., observed glaciers are predominantly large and strongly dynamic) in the sample of observed glaciers. It is estimated empirically based on climatological and size differences between observed and unobserved glaciers.

These four terms are combined as:

$$\sigma_{\bar{h}} = \sqrt{\left(\frac{\sigma_C}{\sqrt{n_{obs}}}\right)^2 + \mu_C^2 + \sigma_{obs}^2 + \sigma_{repr}^2}. \tag{6}$$

For RGI regions without point observation of ice thickness, we assume that the percent error is double that of the regions which their parameters are based on. All individual error terms contributing to $\sigma_V$ are listed in Table S2. Full details and graphical illustrations on the uncertainty quantification are given in the supplement.

To estimate point uncertainty of the modeled bed elevations, we refer to Fig. S1 and the MAE between modeled and observed thicknesses for each RGI region. All observations were used to calibrate the model to optimally constrain the final bed product. To gauge whether the MAEs in Fig. S1 are representative of the true uncertainty, we performed an independent experiment in three well-observed regions (Svalbard, Central Europe and Arctic Canada South) where we randomly selected and withheld 50% of the thickness observations during calibration and used them for validation. We ensured that data from one glacier was entirely in either the calibration or validation dataset. Fig. S7 shows the example of Aletsch glacier and surrounding glaciers in the Swiss Alps modeled without using their thickness observations. In Svalbard, the MAE on the withheld data is 83.4 m, in Central Europe 33.3 m, and in Arctic Canada South 109.6 m. In comparison, the MAEs of the all-data calibration (Fig. S1) in these three regions are 77.4 m, 31.4 m and 98.2 m, respectively. These latter numbers are, on average, 8.5% lower than the MAEs on withheld data, indicating a slight performance degradation on unobserved glaciers. Meanwhile, the MAEs on the observations used for calibration in the withholding experiment are 73.3 m, 28.7 m, and 86.9 m in Svalbard, Central Europe and Arctic Canada South, respectively. Averaged across the three regions, the MAEs of the all-data calibration (Fig. S1) are 8.5% larger than these numbers. Pooling withheld and non-withheld observations of the withholding experiment together, the MAEs in the three regions are 79.8 m, 31.0 m, and 98.4 m, respectively. The all-data MAEs are on average 0.7% smaller than those numbers, i.e. essentially equal. This withholding experiment indicates a modest degree of data dependence due to the slight increase in MAE on withheld data relative to the all-data calibration, while MAEs on non-withheld data are correspondingly lower. The negligible difference between the pooled MAEs of the withholding experiment and the all-data MAEs indicates that the latter values as shown in Fig. S1 provide a robust first-order estimate of regional point bed elevation uncertainty, although in data-sparse regions, the error in unobserved locations may be moderately underestimated.

## Lake mapping

Potential future lakes are mapped by running a sink-fill algorithm[78] implemented in RichDEM[79] on the output bed topography inside the glacier outlines. Lake volumes are calculated as the difference between the filled and the original elevations times the grid area. Lakes with an area < 0.05 km² and cells with a depth < 5 m are removed to avoid spurious effects of uncertain small-scale topography. Not applying these constraints increases lake volume by only 2%, although lake area would be 50,914 km², i.e. 25% larger. This would raise the potential future percentage of lake cover of the presently glacierized landscape to 7% (instead of 5.8%). Lakes located within a buffer of twice the grid resolution from the ocean are also removed as they are unlikely to remain disconnected on a relevant time scale. For estimating the SLE of the volume retained by the lakes, only the lake volume above sea level is considered, as volume below sea level presently filled with glacier ice already today acts as a sink for water otherwise contributing to sea-level rise.

## Ice-volume to sea level equivalent conversion

Glacier volumes are converted to SLE according to

$$SLE = \frac{1}{A_{ocean}} \frac{\rho_{ice}}{\rho_{ocean}} \int_{\Omega} \max\left(0, h - \frac{\rho_{ocean}}{\rho_{ice}} D\right) dA \qquad (7)$$

where the density of glacier ice $\rho_{ice}$ = 917 kg m$^{-3}$, the ocean water density $\rho_{ocean}$ = 1028 kg m$^{-3}$, the global ocean area $A_{ocean}$ = 3.618 × 10$^8$ km$^2$, $\Omega$ is the glacier-covered domain, and $D = \max(-B, 0)$ is water depth. Eq. (7) accounts for ice below flotation that does not contribute to sea level rise if melted.

## Data availability

All modeled bed topographies, ice thicknesses and lakes are available as GeoTIFFs via https://doi.org/10.6084/m9.figshare.29940932.

## Code availability

IGM v2.2.1 is available from https://github.com/instructed-glacier-model/igm. A new IGM module implementing the inversion, the Bayesian calibration workflow and the code for mapping lakes is available via https://github.com/hahohe1892/Frank_thk_inv.git[80].

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

## Acknowledgements

We acknowledge Fabien Maussion for his implementation of the OGGM shop in IGM, which is used extensively here. T.F. and W.v.P acknowledge funding from the Swedish Research Council (grant No. 2020-04319). W.v.P. was further supported by the Swedish National Space Agency (project No. 189/18 and 2024-00241). D.R. acknowledges support from the National Aeronautics and Space Administration (NASA, grants 80NSSC20K1296, 80NSSC24K1638, 80NSSC24K1530). R.H. was supported by grants from the Norwegian Research Council (324131 GLAC-MOD), ERC-2022-ADG 101096057 GLACMASS, and NASA (80NSSC20K1296 and 80NSSC20K1595). The computations were enabled by resources provided by the National Academic Infrastructure for Supercomputing in Sweden (NAISS) at Chalmers, partially funded by the Swedish Research Council through grant agreement No. 2022-06725.

## Author contributions

T.F. and W.v.P. conceived and designed the study with the help of R.H. T.F. implemented the inversion method used here as a module within IGM, which was developed by G.J. T.F. furthermore designed and conducted the simulations with the help of WvP and technical guidance from G.J. D.R. provided mass balance data. T.F. wrote the manuscript, produced the figures and carried out the data analysis with significant inputs from W.v.P., R.H., G.J. and D.R.

## Funding

## Competing interests

The authors declare no competing interests.
