## [Transparent Peer Review file · Nature Communications]

Global glacier-free topography reveals large potential for future lakes in presently ice-covered terrain

Corresponding Author: Dr Thomas Frank

Version 0:

Reviewer comments:

Reviewer #1

(Remarks to the Author)

GENERAL ASSESSMENT

This manuscript presents a well-written and timely study on the global reconstruction of subglacial topography beneath glaciers outside the Greenland and Antarctic ice sheets. The authors introduce the TOPO-DE dataset, derived through inversion using the Instructed Glacier Model (IGM)—a 3D higher-order ice flow model accelerated by GPU and deep learning. This approach is theoretically an improvement over existing datasets such as those by Farinotti et al. (2019) and Millan et al. (2022).

Based on TOPO-DE, the authors derive estimates of potential future lakes and provide updated global and regional glacier volume data—similar in scope to the aforementioned studies. As such, the novelty of the paper lies primarily in the new modeling methodology and the inclusion of lake formation estimates.

The topic is highly relevant to the cryosphere and broader Earth system science communities. If TOPO-DE can be shown to outperform existing datasets in a robust and quantitative way, it has the potential to become a foundational resource for future research on glacier retreat, lake formation, and sea-level rise.

The paper is methodologically ambitious and addresses a critical gap in our understanding of glacier bed topography and its implications. The writing is clear, the figures are informative, and the data availability is commendable.

SIGNIFICANCE AND NOVELTY

- The study presents the first physically consistent global reconstruction of glacier beds using a higher-order ice flow model, overcoming limitations of shallow ice approximation (SIA)-based approaches. This is a significant methodological step, but the quality of the resulting dataset requires more thorough evaluation.
- The estimation of over 56,000 potential future lakes with a total volume of 3,138 km³—covering ~6% of the newly deglaciated terrain—is a novel and valuable contribution.
- The refined global glacier volume ($149.41 \pm 29.00 \times 10^3$ km³), corresponding to 316 ± 61 mm sea-level equivalent, aligns well with previous studies. While not novel, it serves as an additional confirmation of existing estimates.
- The TOPO-DE dataset is openly available and has potential utility across disciplines.

The methodology is well-documented and includes:

- A physically-based inversion framework leveraging mass conservation and the computational strengths of IGM (GPU acceleration and deep learning).
- Regional calibration using Bayesian optimization against >3.8 million thickness observations. However, it should be noted that these observations appear to be used for both calibration and validation, raising concerns about the independence of the validation data.

MAJOR COMMENTS

However, two major concerns should be addressed to strengthen the manuscript:

- Major Comment 1: Need for Quantitative Comparison with Existing Products

The TOPO-DE dataset is presented as a methodological improvement over existing products by Farinotti et al. and Millan et al.. While the theoretical advantages of using a 3D higher-order model are well-articulated, the manuscript lacks a robust quantitative comparison to demonstrate that TOPO-DE is superior in practice.

Currently, the improvements are described in qualitative terms (e.g., “the realism of glacier beds is visibly improved”), while

performance metrics against observed thicknesses are said to show “similar skill as previous studies.” This weakens the central claim of the paper.

To substantiate the value of TOPO-DE, the authors should:

- * Include quantitative comparisons (e.g., RMSE, bias, spatial consistency) across representative regions and glacier types.
- * Clarify whether the improved realism leads to better predictive performance or reduced uncertainty in downstream applications, rather than being primarily a visual improvement.

- Major Comment 2: Lack of Independent Validation

The use of all available point thickness observations for calibration means that these data cannot be considered independent for validation. This is a methodological limitation that should be addressed by adopting a more standard machine learning framework, with separate training and testing datasets.

The authors mention an experiment (L278–283) where 50% of the data were withheld for validation. However, it is unclear how this resulted in a lower MAE on the withheld data compared to the full-data calibration. This contradicts expectations of overfitting and suggests either a misinterpretation or a methodological issue.

To clarify and improve this aspect, the authors should:

- * Clearly describe the data split and validation protocol.
- * Report performance metrics separately for training and testing sets.
- * Explain the unexpected improvement in testing MAE, or revise the interpretation accordingly. Or clarify my misunderstanding of their approach.

SPECIFIC COMMENTS

L49-54: it is clear how some of the methodological limitations are addressed in this study (e.g. 1D vs 3D), but for other limitations it is not very clear. E.g. how are difficulties in constraining ice velocities on slow-flowing glaciers addressed in this study?

L57: The phrase “an innovative bed inversion” should be clarified—what exactly is innovative?

L68: “We compile” should be revised to “We have compiled.” It would also be helpful to clarify whether this compiled dataset is publicly available.

L74: “Run for at least 2000 years” — it is unclear where this duration is defined or justified in the methodology.

L80: “3,800,000” — how many of these points are used for training/calibration versus testing?

L95 “similar skill” this is very subjective and quantitative. What is similar? Which metrics? This quantitative difference is key for the paper and its motivation but is only subjectively mentioned

L96 “is visibly improved in many cases” → this is very subjective and vague? How do we know it is better? What are many cases (how many)? What are the quantitative criteria to say that it is better

L110 “the first global product derived from a method capable of producing physically plausible bed shapes for all glaciers on Earth, in contrast to existing SIA-based methods.” This is just pure promo talk and is not shown. Yes, IGM is more complex than SIA methods but it is also based on CNNs and these are far from “physically driven”. Therefore, this sentence needs potential down-toning and the added value should be quantitatively shown, not be assumed based on theoretical assumptions (e.g. a badly calibrated 3D model can be worse than a simple but good 1D model).

L141: The methodological difference regarding averaging model outputs in previous studies should be explained more clearly. Are individual Farinotti outputs better? Why is averaging not needed in TOPO-DE?

L277-283 “we perform an independent experiment where 50% of them are withheld for validation in three well-observed regions” Not clear what is meant here and how it is possible that a model trained with less data and more independent test data can be better than the original model with all data and potential of strong overfitting

Fig S1: Excellent figure.

Fig S2: Consider removing rainbow colormaps, which are known to be misleading (see <https://www.nature.com/articles/s41467-020-19160-7>).

(Remarks on code availability)

Reviewer #2

(Remarks to the Author)

This paper presents a significant piece of research that does more than the title give away. The authors use a higher-order flow model to produce a new sub-ice bed topography map for the globe, modelling the topography of each glacier (or coherent group of glaciers) at between 400 and 100 m resolution. The authors assume some basal conditions relating to friction and flow and apply a numerical inversion that is, where possible, observationally constrained with ice thickness measurements. The model itself is one that is proving to be highly efficient, using deep learning to accelerate its performance and make the application to all glaciers of the globe possible. The result of the work, a new open access bed topography for the globe is a significant output that is likely to be of use to all types of earth scientists, from glaciologists to geologists. Furthermore, the outcome of this work and dataset for this paper is twofold: first the presentation of (and discussion of) new ice volume (and sea level equivalents) estimates for each glacierized region. Second is the development of an understanding of where lakes would appear were all the glaciers to disappear. Both are significant findings and the paper here presents the lakes as the first major outcome and the sea level/ice volume understanding as the second science outcome.

In general the manuscript is well written, although in places there is some vagueness in meaning and in places additional detail would be useful. The manuscript is doing a good job of representing a large volume of work and processing, but there

is scope to enhance description (quantitative and qualitative) of the outcomes. The figures are of good quality – very clear and easy to follow – including in the supplementary files.

Overall, I think the bed topography data (and ice thickness data) are robust and will be highly useful. I think that the use of them to compute where lakes are present is also valid. There could be some indication in the spatial data of the uncertainty associated with topography in particular areas (see below).

The data and methodology are well described, with excellent detail in the supplementary documents. There are some areas where extra detail/figures could be further added (see below).

The analytical approach in terms of generating the dataset seems very strong overall – it is well considered. The analysis of the landscapes themselves is much lighter touch – with lakes being picked out as the key component, but even then, although the lakes are presented as a dataset and the lake volumes are presented in a main figure, there is not much spatial analysis of the lake characteristics (e.g. sizes, numbers etc per region or per elevation band etc.). If management and hazard is to be an important part of the paper then it would seem important to analyse these areas in more detail.

Suggested improvements: (Key areas):

I found myself thinking that the major scientific advance presented here is not really the lakes that may form when the ice disappears but it is the production of the glacier bed topography and calculation of subsequent ice volumes and sea level equivalent that is the significant advance. Moreover, within that, the significant advance is the form of that topography, which at present is only lightly touched upon. The lake component is part of that and is an interesting outcome too of course, but I was surprised that the title of the paper is selling this as the key point, and was surprised that if this is the key point then it is not explored in more depth. In the end, I think that more could be made of knowing the bed and ice volume in the first place and the new bed database is definitely a significant advance there. A secondary finding is where lakes would appear if deglaciation were to happen. I think that there are two options for adjusting the paper depending on what the authors prefer to be the big point: 1) Make the topography and ice volume estimates the big point by restructuring to talk of these first and then bring in the lake point as part of a wider discussion about the form of the landscape in general. Or 2) If the lakes are to be the main point, then I think they need explored more in depth. Why they are where they are, more on their form and location within the glacierized terrains, and if the impact of them is thought to be significant then this needs explored more in depth too. On that latter point, GLOFs are pointed out as a key hazard potentially associated with these based on comparing them to modern proglacial lakes. However, many hazardous proglacial lakes are moraine dammed, whereas the lakes being mapped here are really bedrock ones. So is the hazard potential of the future lakes really that significant? The hazard potential should be explored more if the lakes component is to be sold as the key finding.

If lakes are to be a key point in the paper, it would be good to see more text focussed on exploring where they are, what types of topography they are in, that this means for a range of things that you said a lack of knowledge is limiting management of. There is more that could be said on the lakes IF that is the thing you want to really show off here.

The 'realistic' nature of the topography is a key point being made several times in the manuscript. I agree that it appears 'realistic' in the examples I can see but I think it would be helpful to go into more detail/depth about what makes it realistic (and where it is not realistic – provide some detail about where and why such issues occur). On this point of understanding the realistic nature of the topography, it would also be useful to include more figures of where key types of features are modelled as being present – e.g. to back up the point on the examples of these features being present. This is particularly important given that many of them would then be home to the lakes subsequently.

Associated with the realistic nature of the landscape, there is a good quantification to uncertainty outlined in the extended data, but in the main text of the manuscript the uncertainties are not really brought in where quantities are being mentioned. E.g. where ice volumes are outlined, what happens to those when uncertainty is accounted for? Its in the table but not really brought in in the text. How does that uncertainty impact the comparison of volumes against the work from other papers? It would be useful to have the associated assessment of uncertainty/accuracy included in the database. Moreover, more uncertainty occurs under particular data/modelling conditions, and so where such conditions occur you might have a grid that highlights you have less confidence in that area.

Suggested improvements (minor: by line number):

Title: The landscapes are not currently 'lake-rich'. They only would be if/when the glaciers disappear. Thus the title, which implies the lakes exist under the earths glacier is not quite appropriate. One option would be to use a more topographic term if you wish to talk about what is under the glaciers now (e.g. overdeepenings?) or alter the title to reflect the fact that the landscapes would hold lakes only once the glaciers have gone.

11: 'Retreating glaciers profoundly reshape landscapes. This could be confused with the idea of erosion under glaciers which also reshape landscapes. Here I assume that you are trying to say that the landscape looks a lot different once the glaciers disappear? If so, perhaps rephrase this sentence.

18: The lake density calculation mentioned, which is conducted later in the paper, needs expanded upon at some stage in the paper. In particular, how does this compare to previously glaciated regions that are now ice free? E.g. is the number of lakes in your future deglaciated terrain similar to that of the most recently deglaciated terrain? At the moment, I feel like you compare it to all ice free terrain, whether previously glaciated or not. This is fine, but I also think you need the comparison against terrain that has been glaciated as this would be an interesting sense-check on the realistic nature of your dataset.

15-18 vs. 19-20: If the paper is restructured to put the ice volumes etc. first, then adjust the order of these groups of sentences.

35: 'new'. Its not really 'new' landscape - its already there - just buried. So delete 'new'?

36-37: grammatically awkward. The acceleration point could be added to end of previous sentence. Then the poorly known point could be added to the start of the next sentence where you say its poorly known and why that is important.

37: 'severe': is it severe? Need to back this up with an indication of why its severe. E.g. what are the impacts of not being able to manage these things? Thy is it a limitation? If the lakes are to be the key point of this paper, then much more needs to be made of the importance of knowing where they may be in the future, and also what their importance may be. In addition – later – much more would need to be done in exploring where they are regionally and within regions and within what types of topography.

39: 'to mitigate potential hazards'. Such as what?

40-43: This feels structurally awkward. Why is ice behaviour/volume not the first point being made? I say this because the exposed topography would come after the ice disappears and so logically would feel like that should be the way it is written.

47-48: this needs expanded on slightly - are the discrepancies and uncertainties different between the Farinotti paper and the Millan paper? If so, say so directly. If not, indicate how discrepancies are known etc.

54: perhaps slight mention of what is unrealistic about them? How is it known they are unrealistic?

87: great. These are expected, but sometimes they are expected in particular places within a glacier catchment. Therefore, it may be worth mentioning where you tend to find such features. E.g. are the cirques always found at the heads of the valleys? Are the overdeepenings found in most trunks or only where confluence of glacier flow occurs? Basically, can you describe more about where such features are found within the landscapes? Do they also tend to be found in particular quantities in particular regions? Expand a bit if possible.

93: Is isostatic unloading from ice removal accounted for in TOPO-DE? I realise it may not make a huge difference to local features such as lakes but in some areas it may influence things like the shoreline or the overall regional gradients in a landscape. Accounting for this would be important to maximise the usefulness of TOPO-DE. Alternatively, a statement indicating it is not accounted for would be appropriate to add in the methods section.

107-108: I think if inaccuracies are going to be mentioned then it would be more open to then provide a bit of clearer information on that - perhaps in the supplementary information section. And with some example figures to go with text describing the issues.

110-111: It would seem like this could be a good idea for a supplementary figure here to show a figure of topography in an already well measured area vs. your new inverted topography (in a version you create without control data from that particular location). A figure would be particularly useful given you use some supplementary figures to critique (rightly-so) the previously existing topographic reconstructions – so therefore an appropriate exploration/critique of your own dataset should be presented as some text with potentially a supplementary figure.

128-130: perhaps link this back to specific types of features in that landscape? E.g. overdeepenings are larger under the large systems vs. cirques being more dominant in the steeper smaller systems?

132-134: what is average lake size and total lake counts in such areas? And average lake volumes? Suggest quantifying in the text.

142-143: If you are going to talk about GLOF then one key point is that many recent GLOF events have been in moraine-dammed proglacial lakes. Presumably, however, you are making the assumption that the deglaciated bed is bedrock and thus that the lakes are impounded not by moraine but by rock. As a result, I wonder if you are over-emphasising the potential GLOF risk in a deglaciated environment? Perhaps the mode of impounding could be discussed? Discuss the GLOF risk in more detail overall if lakes are a key point of the paper.

145: 'Site specific opportunities and constraints must be assessed carefully'. This feels rather vague. Can you be more specific?

153: Perhaps worth indicating why (e.g. they did not recover the bed in appropriate detail or their assumptions were too limiting).

155-158: While reflecting on differences, it would be useful to indicate why such differences occur. For example, is it because you predict overdeepenings better, or thinner/thicker ice in valleys compared to previous estimates etc? In other words, where in the landscape is the extra ice found in comparison to reference 13? Just thicker overall? Or more ice in valleys due to overdeepened topography? Explore this a bit.

173-174: This is a tagline that is used in the paper a couple of times, but its not really explored much apart from the GLOF component. Is more exploration possible?

175-181: This is where you could do more exploration of the above. But in any case, these sentences feel rather vague in terms of what the new database is really going to be useful for. I think a better case for the impacts of this work could be made. You do pick up on the key point to future global modelling, but in an overly-technical manner. This paragraph needs to be made more accessible for a wide readership as is the case in Nature Communications.

208: is a particular type of friction/sliding therefore assumed?

212: There is uncertainty in ITS_LIVE. How is that translated through to the inversion?

224-225: this feels vague. What is meant by neighbouring regions?

243: Would that tend towards undershoots or overshoots? Does it have a tendency to put particular shapes into the topography where such interpolation is used?

277-280: A map-based figure to show how the new database compares with the measured thicknesses in one of these areas could be a useful one. It would allow better understanding of how well the approach does in estimating the thickness and bed.

Extended Data Table 2: This table shows how much RGI appears to differently estimate SLE compared to your work. In the text at present, you give the example of the subantarctic which is pretty much the only one where RGI and your estimate are close. Therefore I suggest an additional sentence in the text which clearly indicates that RGI has a strong tendency to significantly overestimate or underestimate compared to your work.

On that point, you quote the mean differences in parentheses here between your work and others. The means are not provided for RGI - it would be interesting to have that. But even then, the means are not necessarily helpful because there are many + or - values that are quite large depending on region. Would it be useful to separate our mean underestimations and mean overestimations? And the same for Millan. Just a thought - might pull out whether the underestimates or the overestimates are the key differences in the datasets.

References: These seem to cover the important bases.

(Remarks on code availability)

I have checked the code is available and that the instructions are clear (all seems good), but I have not downloaded and run the code.

Version 1:

Reviewer comments:

Reviewer #2

(Remarks to the Author)

This paper has undergone substantial revisions since its first iteration. There are significant expansions of both the uncertainties and descriptions of the topographies, and the description of the lakes dataset that is derived from the bed topography. The authors have clearly taken a lot of time and effort to respond very carefully to the comments that both myself and the other reviewer made and in doing so have delivered well-detailed additional text that helps with both the methodological understanding and the key findings. The adjustments to figures, for example the switch to a different ice cap as an example of the model capabilities, are useful and appropriate. The addition of new figures is also beneficial and these are particularly helpful in understanding how the lakes are distributed by elevation - the text linked to this is very informative too. The authors push back on a couple of minor points - I don't have any problems with this - the authors justify these well, especially as a couple are because I missed things in my initial review - my apologies for that.

Overall, I do not see any need for further adjustment aside from one tiny minor comment that if there happens to be a name for the Greenlandic ice cap in figure 1 then it would be appropriate to name it, or to name the specific region that it lies within (use indigenous names if possible).

Overall, the careful and detailed responses and additions mean that I would be very happy to see this paper published in its current form.

(Remarks on code availability)

I have not reviewed the code itself, because I would likely not be expert enough to use it. But I have reviewed the data that comes with the paper as these form the core of the outputs - these are all appropriate, and I see that they will be made available via repositories alongside publication.

Global glacier-free topography reveals large potential for future lakes in presently ice-covered terrain - reviewer response

February 2026

We thank the reviewers for their constructive comments which we have addressed below. The comments are in italics, our responses are regular font and added or changed sentences in the revised manuscript are marked in blue.

1 Adjustments additional to reviewer comments

In addition to addressing all reviewer comments, the following changes were made to the manuscript:

- We replaced Vatnajökull in Fig. 1 with an ice cap from Greenland since we realized that an ice cap from Iceland was not the ideal choice for showcasing given that this region is comparatively poorly constrained due to the absence of thickness observations in the GlaThiDa and due to various processes specific to this region that are not well represented in the model (e.g. calderas with very large basal melt rates).
- The manuscript was restructured to comply with Nature Communications formatting guidelines. This mainly included removing the "Extended Data" section and placing items previously found there either in the main text (previous Extended Data Table 1) or in the Supplementary Material (all other items). Furthermore, the abstract was shortened to comply with the 150 words limit of Nature Communications.
- In Figs. S3a, S4a (formerly Extended Data Fig. 2a,3a) glacier outlines in light-gray were added to highlight the boundaries of individual outlet glaciers.
- Minor textual changes in a few places were done to improve flow. These changes can be seen in the supplied difference file.

Figure 1: Examples of model output for Fedchenko glacier in Central Asia (central coordinates: $72.23^{\circ}\text{E } 38.80^{\circ}\text{N}$) and an ice cap in Greenland ($26.78^{\circ}\text{W } 71.22^{\circ}\text{N}$). a,c) Modeled ice thickness draped over present-day surface topography; b,d) glacier-free topography with future lakes. Topographies in all plots are shown as hillshades.

2 Reviewer 1

2.1 General assessment

This manuscript presents a well-written and timely study on the global reconstruction of subglacial topography beneath glaciers outside the Greenland and Antarctic ice sheets. The authors introduce the TOPO-DE dataset, derived through inversion using the Instructed Glacier Model (IGM)—a 3D higher-order ice flow model accelerated by GPU and deep learning. This approach is theoretically an improvement over existing datasets such as those by Farinotti et al. (2019) and Millan et al. (2022). Based on TOPO-DE, the authors derive estimates of potential future lakes and provide updated global and regional glacier volume data—similar in scope to the aforementioned studies. As such, the novelty of the paper lies primarily in the new modeling methodology and the inclusion of lake formation estimates. The topic is highly relevant to the cryosphere and broader Earth system science communities. If TOPO-DE can be shown to outperform existing datasets in a robust and quantitative way, it has the potential to become a foundational resource for future research on glacier retreat, lake formation, and sea-level rise. The paper is methodologically ambitious and addresses a critical gap in our understanding of glacier bed topography and its implications. The writing is clear, the figures are informative, and the data availability is commendable.

2.2 Significance and novelty

- The study presents the first physically consistent global reconstruction of glacier beds using a higher-order ice flow model, overcoming limitations of shallow ice approximation (SIA)-based approaches. This is a significant methodological step, but the quality of the resulting dataset requires more thorough evaluation.
- The estimation of over 56,000 potential future lakes with a total volume of $3,138 \text{ km}^3$ —covering 6% of the newly deglaciated terrain—is a novel and valuable contribution.
- The refined global glacier volume ($149.41 \pm 29.00 \times 10^3 \text{ km}^3$), corresponding to $316 \pm 61 \text{ mm}$ sea-level equivalent, aligns well with previous studies. While not novel, it serves as an additional confirmation of existing estimates.
- The TOPO-DE dataset is openly available and has potential utility across disciplines.

The methodology is well-documented and includes:

- A physically-based inversion framework leveraging mass conservation and the computational strengths of IGM (GPU acceleration and deep learning).
- Regional calibration using Bayesian optimization against >3.8 million thickness observations. However, it should be noted that these observations appear to be used for both calibration and validation, raising concerns about the independence of the validation data.

2.3 Major comments

2.3.1 Major Comment 1: Need for Quantitative Comparison with Existing Products

The TOPO-DE dataset is presented as a methodological improvement over existing products by Farinotti et al. and Millan et al.. While the theoretical advantages of using a 3D higher-order model are well-articulated, the manuscript lacks a robust quantitative comparison to demonstrate that TOPO-DE is superior in practice. Currently, the improvements are described in qualitative terms (e.g., “the realism of glacier beds is visibly improved”), while performance metrics against observed thicknesses are said to show “similar skill as previous studies.” This weakens the central claim of the paper. To substantiate the value of TOPO-DE, the authors should:

- Include quantitative comparisons (e.g., RMSE, bias, spatial consistency) across representative regions and glacier types.
- Clarify whether the improved realism leads to better predictive performance or reduced uncertainty in downstream applications, rather than being primarily a visual improvement.

RESPONSE: We addressed these points with the following additional analysis:

- **Error metrics against GlaThiDa**

We calculated the MAE against GlaThiDa observations for all global studies, showing that our product has the lowest error globally. However, while this can serve as a performance indication, it cannot serve as a proof for superiority of TOPO-DE given that all GlaThiDa observations were used for calibration, in this study and in the previous studies. In the main text, the following sentence was added, alongside a new supplementary figure (Fig. 2): “Error metrics against observed thicknesses indicate improved skill compared to previous studies (Fig. S1, S2), although these metrics should be interpreted with caution because a shared, independent validation data set across studies is lacking (Supplementary Discussion).”

The added supplementary discussion giving the error metrics for all studies and detailing why a quantitative ranking of model performance should be interpreted with care reads:

”Quantitative comparison between thickness products

To quantitatively compare the thickness products from this study, [1] and [2] we calculate their agreement with GlaThiDa observations [3, 4] at the global scale. We obtain MAEs of 122 m (ME: 33 m) for [1], 106 m (ME: 11 m) for [2], and 94 m (ME: 12 m) for this study (Fig. S2), considering for each study only those glaciers for which it produced an output.

However, these numbers need to be interpreted with care due to the lack of a shared pool of independent validation data. A definitive assessment of model performance requires observations that were not used for model calibration. In practice, all studies used the full set of available GlaThiDa observations for that purpose. Moreover, only few thickness measurements have become available in recent years that are not yet included in

the GlaThiDa [3], and these new data remain too sparse to support a comprehensive global evaluation. This precludes a fully independent quantitative assessment of model performance.

Meaningful intercomparison in the absence of independent validation data depends on whether the degree to which different calibration approaches allow models to generalize from observed to unobserved locations is comparable. This generalization skill is closely related to the effective number of degrees of freedom in the calibration. Approaches based on the adjustment of a limited set of region-wide parameters impose strong structural constraints and therefore favor generalization, whereas methods that locally assimilate individual measurements introduce substantially more degrees of freedom, enabling very close agreement with observations but increasing the risk of overfitting.

Under the assumption that different studies generalized the information from observed to unobserved locations in a broadly comparable manner, a ranking based on performance at observed locations can be informative. This study and the work by [1] and [2] all relied predominantly on region-wide parameter calibration with only a few exceptions, such as one model in the ensemble approach by [1] that locally assimilated thickness observations [5]. Given the otherwise similar calibration strategies we find the above metrics indicative of model performance, though not fully conclusive in the absence of independent validation data.”

- **Performance for different thickness classes**

We analyzed the performance of the different products for different thickness classes. To that end, we calculated the difference between observed and modeled thicknesses for ten thickness bins, with each bin containing 10% of the observations (Fig. 3). Note that due to differences in the spatial coverage between the different products, the bins are not exactly identical (however, the same analysis with identical bins not shown here yields the same conclusions). We find that [1] and [2] both systematically and strongly overestimate small thicknesses and underestimate large ones. For our product, the same trend is visible to a much lesser degree. The following was added to the main text, alongside Fig. 3:

”Importantly, TOPO-DE substantially reduces systematic biases found in earlier thickness estimates which tended to overestimate thin and underestimate thick ice (Fig. 2). This has important implications for accurate glacier projections: With TOPO-DE, thinner glaciers with smaller initial thickness will disappear faster under future warming compared to using previous (greater) thickness estimates. Thicker glaciers, by contrast, are expected to survive longer. These biases would naturally also affect the mapping of potential future lakes. For example, unrealistically smooth beds may be caused by underestimating thickness variability within individual glaciers, which could imply that the potential for lakes is underestimated.”

- **Identifying artifacts in existing products on a large scale**

We extended the analysis on the unrealistic bed shapes seen in [1] and [2] (Extended Data Fig. 2 and 3 in the original manuscript, now Figs. S3, S4) by computing the bed roughness R_B . We show R_B fields for the same two example glaciers as in Figs. S3, S4 (Figs. 4, 5) and link the patterns found to the global scale. Specifically, we added the following analysis to the Supplementary Discussion:

”In Figs. S3, S4, we show examples of unrealistic bed topography found in the products by [1] and [2]. These manifest predominantly as ”walls” between outlet glaciers of glacier complexes and highly irregular bed topography on small glaciers. To quantitatively describe these artifacts, we compute the bed roughness as the root squared slope

$$R_B = \sqrt{\left(\frac{dB}{dx}\right)^2 + \left(\frac{dB}{dy}\right)^2} \quad (1)$$

for the shown examples (Figs. S12, S13) and all glaciers globally, for this study and existing products [1, 2]. In Figs. S12c and S13c, it is evident that the ”walls” between outlet glaciers show up as very large R_B values, due to the sudden jumps in bed height. These many large values result in a spike in [1]’s R_B histogram for $R_B \geq 0.8 \text{ m m}^{-1}$ (Figs. S12a and S13a). In Fig. S13d, we find that the product by [2] exhibits elevated R_B values across most of the domain where this study and [1] show a smooth bed. This can be traced back to the overall noisy bed appearance seen in Fig. S4d. Going beyond these two examples to the global scale, we find that the product by [1] has the largest mean bed roughness for large glaciers ($>10 \text{ km}^2$) whereas [2] is the roughest for small ($\leq 10 \text{ km}^2$) glaciers. We interpret this to reflect the patterns seen in the examples discussed here (Figs. S12 and S13): For large glacier complexes with many outlet glaciers, the artificial ”walls” produced by [1] make the bed unrealistically rough. Meanwhile, for small glaciers, the small-scale bed noise introduced by large relative uncertainties in mapping slow ice flow velocities results in too-rough beds in [2]. This quantitative analysis demonstrates that the artifacts seen in Figs. S3, S4, which TOPO-DE is free of, impact the realism of the existing bed products on a large scale. Note that the elevated roughness of

the products by [1] and [2] described here does not contradict possibly too smooth bed shapes indicated by the thickness-dependent biases seen in Fig. 2: the roughness quantifies bed variations from one grid point to the next whereas the previously described biases concern patterns on a larger scale, such as the systematic underestimation of the depth of subglacial troughs and the height of bed peaks.”

- **Geological patterns not seen in Farinotti and Millan**

We added the bed maps of [1] and [2] to Extended Data Fig. 4 (original manuscript) which shows how TOPO-DE can aid geological mapping of currently glacierized terrain (Fig. 6). The linear features seen in TOPO-DE which we strongly believe to be real due to their alignment with the stratigraphy outside the glacier are barely [1] or not [2] visible in the other products. This provides another line of evidence that TOPO-DE is superior in terms of physical realism. Apart from updating Fig. 6, the following was changed in the main text: “For instance, a sequence of resistant and weak rocks mapped outside a glacier in the Canadian Arctic [6] can now be traced in detail beneath the ice, whereas the same formations were weakly expressed or absent in previous products (Fig. S5). This highlights the potential of TOPO-DE to aid geological mapping of currently ice-covered landscapes.”

In summary, here we provide several lines of evidence why TOPO-DE is physically more plausible than existing products. In addition to these points, we again want to stress that there are clear methodological advantages of our approach as to why this should be the case. Although impossible to prove by validation against independent data, we argue that together this provides sufficient evidence that TOPO-DE is superior to existing products.

Figure 2: Modeled versus observed ice thicknesses aggregated globally for this study and existing products by [7] and [2] with metrics (MAE: Mean Absolute Error, ME: Mean Error, unit: m). Point color indicates point density, with darker colors showing a larger overlap of points. Red dashed line indicates 1:1 line.

Figure 3: Boxplot of the model-observation misfit for ten thickness classes, each containing 10% of the data, for this product and the thickness products by [1] and [2]. Horizontal orange lines represent the median and green triangles the mean misfit. Note that the thickness classes differ slightly due to different spatial coverage of the products. The same analysis with uniform classes yields the same patterns (not shown).

Figure 4: Comparison of bed roughness R_B with existing products for a glacier complex in Greenland (central coordinates of figure: 26.78°W 71.22°N). a) R_B histogram over the glaciarized area for this study (blue), [7] (yellow) and [2] (green), with dashed lines indicating median values. b) Map of R_B for this study. c) Map of R_B for [7] with large values above 0.8 m m^{-1} found at artificial walls between outlet glaciers, which are also reflected in the histogram. d) Map of R_B for [2].

Figure 5: Comparison of bed roughness R_B with existing products for a glacier complex in Canada (central coordinates of figure: 69.70°W 69.39°N). a) R_B histogram over the glacialized area for this study (blue), [7] (yellow) and [2] (green), with dashed lines indicating median values. b) Map of R_B for this study. c) Map of R_B for [7] with large values found at artificial walls between outlet glaciers, which are also reflected in the histogram. d) Map of R_B for [2] with high abundance of intermediate roughness values throughout the domain due to an overall noisy bed shape, also reflected in the histogram.

Figure 6: Example illustrating how simulated bed topography can aid geological mapping of currently glaciated terrain. a) Surface topography and b) simulated bed topography of Steacie ice cap, Axel Heiberg Island, Northern Canada. c) Close-up view of an area where an alternation of resistant and weak units mapped outside the ice cap [shown in pinkish and green colors, respectively; 6] is seen to continue under the present-day ice cover. This continuation is indicated by the red dashed outlines i) and ii). d and e) Bed topography from Farinotti et al. (2019) and Millan et al. (2022) of the same area. Whereas the longitudinal features in ii) can be seen in both products, too, as they are largely imprinted in ice-free topography, the continuation in i) is not visible in Millan et al. (2022) and barely visible in Farinotti et al. (2019). Red arrows point to example locations where the features seen in c) are much less pronounced or absent in d) and e).

2.3.2 Major Comment 2: Lack of Independent Validation

The use of all available point thickness observations for calibration means that these data cannot be considered independent for validation. This is a methodological limitation that should be addressed by adopting a more standard machine learning framework, with separate training and testing datasets. The authors mention an experiment (L278–283) where 50% of the data were withheld for validation. However, it is unclear how this resulted in a lower MAE on the withheld data compared to the full-data calibration. This contradicts expectations of overfitting and suggests either a misinterpretation or a methodological issue. To clarify and improve this aspect, the authors should:

- *Clearly describe the data split and validation protocol.*
- *Report performance metrics separately for training and testing sets.*
- *Explain the unexpected improvement in testing MAE, or revise the interpretation accordingly. Or clarify my misunderstanding of their approach.*

RESPONSE: We believe this comment largely to be grounded in a misunderstanding, yet we see that we could have been clearer in describing the validation experiment. In contrast to what the reviewer understood, we saw a *higher* MAE on the withheld data compared to the full-data calibration. As written in the original manuscript (l.280): "This [The experiment with withheld observations] **increases** the MAE on the withheld observations compared to the MAE of the all-data calibration, by on average 8.5%".

In the revised manuscript, we have more clearly described how this validation experiment using three regions with many thickness observations was done, entirely separating training and validation performance, and what this says about point thickness uncertainty of TOPO-DE. We acknowledge that apart from the here described validation experiment, we used all thickness observations for calibration and therefore do not have independent validation data (as did previous studies). This was done to ensure that the final bed product is informed by as many observations as possible. Performing a multi-fold training-validation scheme where the data is split into several portions that are used consecutively as training and validation data was not feasible as this would require multiple times the already large computational resources used here. Running the validation experiment in three regions, thus going beyond the efforts of the previous studies which, to our knowledge, did not do such an experiment [1] or did it in one region only [2], is the closest we can get to independent validation.

In the methods section, the validation experiment is now described as follows:

"To estimate point uncertainty of the modeled bed elevations, we refer to Fig. S1 and the MAE between modeled and observed thicknesses for each RGI region. All observations were used to calibrate the model to optimally constrain the final bed product. To gauge whether the MAEs in Fig. S1 are representative of the true uncertainty, we performed an independent experiment in three well-observed regions (Svalbard, Central Europe and Arctic Canada South) where we randomly selected and withheld 50% of the thickness observations during calibration and used them for validation. We ensured that data from one glacier was entirely in either the calibration or validation dataset. Fig. S7 shows the example of Aletsch glacier and surrounding glaciers in the Swiss Alps modeled without using their thickness observations. In Svalbard, the MAE on the withheld data is 83.4 m, in Central Europe 33.3 m, and in Arctic Canada South 109.6 m. In comparison, the MAEs of the all-data calibration (Fig. S1) in these three regions are 77.4 m, 31.4 m and 98.2 m, respectively. These latter numbers are, on average, 8.5% lower than the MAEs on withheld data, indicating a slight performance degradation on unobserved glaciers. Meanwhile, the MAEs on the observations used for calibration in the withholding experiment are 73.3 m, 28.7 m, and 86.9 m in Svalbard, Central Europe and Arctic Canada South, respectively. Averaged across the three regions, the MAEs of the all-data calibration (Fig. S1) are 8.5% larger than these numbers. Pooling withheld and non-withheld observations of the withholding experiment together, the MAEs in the three regions are 79.8 m, 31.0 m, and 98.4 m, respectively. The all-data MAEs are on average 0.7% smaller than those numbers, i.e. essentially equal. This withholding experiment indicates a modest degree of data dependence due to the slight increase in MAE on withheld data relative to the all-data calibration, while MAEs on non-withheld data are correspondingly lower. The negligible difference between the pooled MAEs of the withholding experiment and the all-data MAEs indicates that the latter values as shown in Fig. S1 provide a robust first-order estimate of regional point bed elevation uncertainty, although in data-sparse regions, the error in unobserved locations may be moderately underestimated."

2.4 Specific comments

L49–54: it is clear how some of the methodological limitations are addressed in this study (e.g. 1D vs 3D), but for other limitations it is not very clear. E.g. how are difficulties in constraining ice velocities on slow-flowing glaciers addressed in this study?

RESPONSE: The question raised here (ice velocities on slow-flowing glaciers) is not relevant in our approach as ice velocities were only used in fast-flowing (>100 m yr⁻¹) areas of marine-terminating glaciers. Likewise, walls between individual glaciers of larger glacier complexes are avoided by merging connected glaciers prior to the inversion. This is discussed later, but to indicate already here that we did several additional improvements, we rephrased to: "Here, we overcome these limitations by applying a 3D higher-order ice flow model [8] at global scale, together with a rigorous data processing pipeline."

L57: The phrase "an innovative bed inversion" should be clarified—what exactly is innovative?

RESPONSE: We removed "innovative".

L68: "We compile" should be revised to "We have compiled." It would also be helpful to clarify whether this compiled dataset is publicly available.

RESPONSE: We have rephrased accordingly. We have also added that all input data sets are publicly available, allowing anyone to reproduce our approach. The sentence now reads: "We have compiled a comprehensive dataset from publicly available sources for the > 200,000 glaciers in the globally complete Randolph Glacier Inventory v6.0 [RGI; 9, 10], including digital elevation models, ..."

L74: "Run for at least 2000 years" — it is unclear where this duration is defined or justified in the methodology.

RESPONSE: The justification for that is given in the methods section, L.194-196 (original manuscript): "To achieve convergence, the duration of the simulation should exceed a given glacier's response time. We therefore model at least 2,000, but up to 5,000 yrs per glacier, ...". A reference to the Methods was added here.

L80: "3,800,000" — how many of these points are used for training/calibration versus testing?

RESPONSE: For producing the final bed product, all of these points were used (though nearby observations were averaged when creating discretized input data on the rectangular grid). Please refer to the reply to the major comment for why this was done.

L95: "similar skill" this is very subjective and quantitative. What is similar? Which metrics? This quantitative difference is key for the paper and its motivation but is only subjectively mentioned.

RESPONSE: See response to main comment. Originally, we purposefully remained vague here as there is no independent, quantitative evidence that one product is better than the others. This is grounded not only in the design of this study, but of all three studies that produced global bed maps. With the added discussion we hope to have clarified this point (see above for text changes).

L96: "is visibly improved in many cases" → this is very subjective and vague. How do we know it is better? What are many cases (how many?) What are the quantitative criteria to say that it is better?

RESPONSE: Again, see response to main comment. This specific sentence was reformulated to: "TOPO-DE also visibly improves the realism of glacier beds"

L110: "the first global product derived from a method capable of producing physically plausible bed shapes for all glaciers on Earth, in contrast to existing SIA-based methods." This is just pure promo talk and is not shown. Yes, IGM is more complex than SIA methods but it is also based on CNNs and these are far from "physically driven". Therefore, this sentence needs potential downtoning and the added value should be quantitatively shown, not be assumed based on theoretical assumptions (e.g. a badly calibrated 3D model can be worse than a simple but good 1D model).

RESPONSE: We removed this sentence as we believe that the added discussion clarifies the aspect of improved realism. Meanwhile, we would like to point out that the physics-informed nature of IGM and the specific settings used in this study makes the model very tightly constrained by higher-order ice flow physics with only a small component of (data-driven) emulation.

L141: The methodological difference regarding averaging model outputs in previous studies should be explained more clearly. Are individual Farinotti outputs better? Why is averaging not needed in TOPO-DE?

RESPONSE: Averaging is not needed in TOPO-DE since our methodology is not ensemble-based, in contrast to the Farinotti approach. However, we removed the reference to the ensemble approach here as we believe that the discussion on thickness-dependent biases (see response to main comment) already addresses possible smoothing effects in Farinotti.

L277-283: “we perform an independent experiment where 50% of them are withheld for validation in three well-observed regions.” Not clear what is meant here and how it is possible that a model trained with less data and more independent test data can be better than the original model with all data and potential of strong overfitting.

RESPONSE: See response to major comment. We believe this to be a misunderstanding and have clarified the text accordingly (see above for text changes).

Fig. S1: Excellent figure.

RESPONSE: Thank you! We find that the current color scheme emphasizes mismatches between model output and observations unproportionally due to the dark color given to outliers. We have thus decided to change the color scheme here and for all other scatter plots comparing model output and observations.

Fig. S2: Consider removing rainbow colormaps, which are known to be misleading.

RESPONSE: Thank you for pointing that out, we have changed the color scheme!

3 Reviewer 2

3.1 General

This paper presents a significant piece of research that does more than the title give away. The authors use a higher-order flow model to produce a new sub-ice bed topography map for the globe, modelling the topography of each glaciers (or coherent group of glaciers) at between 400 and 100 m resolution. The authors assume some basal conditions relating to friction and flow and apply a numerical inversion that is, where possible, observationally constrained with ice thickness measurements. The model itself is one that is proving to be highly efficient, using deep learning to accelerate its performance and make the application to all glaciers of the globe possible. The result of the work, a new open access bed topography for the globe is a significant output that is likely to be of use to all types of earth scientists, from glaciologists to geologists. Furthermore, the outcome of this work and dataset for this paper is twofold: first the presentation of (and discussion of) new ice volume (and sea level equivalents) estimates for each glacierized region. Second is the development of an understanding of where lakes would appear were all the glaciers to disappear. Both are significant findings and the paper here presents the lakes as the first major outcome and the sea level/ice volume understanding as the second science outcome. In general the manuscript is well written, although in places there is some vagueness in meaning and in places additional detail would be useful. The manuscript is doing a good job of representing a large volume of work and processing, but there is scope to enhance description (quantitative and qualitative) of the outcomes. The figures are of good quality – very clear and easy to follow – including in the supplementary files. Overall, I think the bed topography data (and ice thickness data) are robust and will be highly useful. I think that the use of them to compute where lakes are present is also valid. There could be some indication in the spatial data of the uncertainty associated with topography in particular areas (see below). The data and methodology are well described, with excellent detail in the supplementary documents. There are some areas where extra detail/figures could be further added (see below). The analytical approach in terms of generating the dataset seems very strong overall – it is well considered. The analysis of the landscapes themselves is much lighter touch – with lakes being picked out as the key component, but even then, although the lakes are presented as a dataset and the lake volumes are presented in a main figure, there is not much spatial analysis of the lake characteristics (e.g. sizes, numbers etc per region or per elevation band etc.). If management and hazard is to be an important part of the paper then it would seem important to analyse these areas in more detail.

3.2 Suggested improvements (key areas)

3.2.1 Part I

I found myself thinking that the major scientific advance presented here is not really the lakes that may form when the ice disappears but it is the production of the glacier bed topography and calculation of subsequent ice volumes and sea level equivalent that is the significant advance. Moreover, within that, the significant advance is the form of that topography, which at present is only lightly touched upon. The lake component is part of that and is an interesting outcome too of course, but I was surprised that the title of the paper is selling this as the key point, and was surprised that if this is the key point then it is not explored in more depth. In the end, I think that more could be made of knowing the bed and ice volume in the first place and the new bed database is definitely a significant advance there. A secondary finding is where lakes would appear if deglaciation were to happen. I think that there are two options for adjusting the paper depending on what the authors prefer to be the big point: 1) Make the topography and ice volume estimates the big point by restructuring to talk of these first and then bring in the lake point as part of a wider discussion about the form of the landscape in general. Or 2) If the lakes are to be the main point, then I think they need explored more in depth. Why they are where they are, more on their form and location within the glacierized terrains, and if the impact of them is thought to be significant then this needs explored more in depth too. On that latter point, GLOFs are pointed out as a key hazard potentially associated with these based on comparing them to modern proglacial lakes. However, many hazardous proglacial lakes are moraine dammed, whereas the lakes being mapped here are really bedrock ones. So is the hazard potential of the future lakes really that significant? The hazard potential should be explored more if the lakes component is to be sold as the key finding. If lakes are to be a key point in the paper, It would be good to see more text focussed on exploring where they are, what types of topography they are in, that this means for a range of things that you said a lack of knowledge is limiting management of. There is more that could be said on the lakes IF that is the thing you want to really show off here.

RESPONSE: We do want to keep a focus on the lakes given that this is the first study to produce such a global dataset, while still striking a balance with the glacier volumes, SLE and bed shape which we very much agree are

highly significant as well. Following the suggestions, we have now substantially expanded the analysis of the lakes. In brief, we added a new Figure (Fig. 7), showing where we find lakes relative to the glacier altitudinal range; we expanded the discussion on lake distribution between different RGI regions; we extended Table 1 (see below) to include more metrics (mean depth, mean area, mean lake surface elevation, mean overlying ice thickness) and we moved that table to the main text; we conducted a thorough literature search and significantly expanded the discussion on the lakes in the context of hazards.

Regarding moraine-dammed lakes, we would like to highlight that these are indeed included in our bed product. We reconstruct the glacier geometry that gave rise to the observed glacier dynamics, making no distinction between a bed consisting of sediments or bedrock. We have now mentioned moraine-dammed lakes in several places of our expanded analysis, clearly signaling to the reader that these are found in TOPO-DE.

To not overload this response letter, we do not reproduce the text of the extended lake analysis here, but refer to the entirely revised subsection "Detecting potential future lakes" in the revised manuscript for all text changes.

Table 1: Statistics of lakes over glacierized area if glacier ice was removed. Global and regional total lake area, number (n), total volume, mean depths, mean area, mean of each lake’s surface elevation (i.e. the elevation of the lake surface if the overlying ice was removed) and the mean overlying ice thickness.

Region	Area (km ²)	n	Volume (km ³)	Mean depth (m)	Mean area (km ²)	Mean lake surface elevation (m)	Mean overlying ice thickness (m)
01-Alaska	7289.2	6385	853.8	117.1	1.12	640	546
02-W Canada & US	314.4	1605	12.3	39.1	0.20	1578	258
03-Arctic Canada N	7246.9	8859	436.1	60.2	0.69	462	352
04-Arctic Canada S	4721.1	5409	317.0	67.1	0.86	452	342
05-Greenland Periphery	5936.9	11518	345.4	58.2	0.52	518	332
06-Iceland	2182.5	797	211.6	96.9	2.73	618	452
07-Svalbard	1907.2	1491	143.9	75.5	1.10	105	364
08-Scandinavia	339.8	1333	15.2	44.7	0.22	1129	229
09-Russian Arctic	3219.6	2756	134.5	41.8	1.17	210	308
10-North Asia	51.2	521	1.8	35.4	0.10	1577	144
11-Central Europe	19.7	143	0.6	31.5	0.14	2337	216
12-Caucasus & Middle East	15.4	84	0.7	42.4	0.15	3034	192
13-Central Asia	817.0	4602	41.3	50.6	0.18	4477	231
14-South Asia W	799.5	2341	51.3	64.2	0.20	4433	319
15-South Asia E	616.4	2466	48.6	78.9	0.25	4755	281
16-Low Latitudes	14.5	144	0.4	30.5	0.10	4942	100
17-Southern Andes	3288.2	3246	408.4	124.2	1.01	559	691
18-New Zealand	34.1	89	3.3	95.7	0.38	916	238
19-Subantarctic & Antarctic Islands	1833.1	2870	111.4	60.7	0.59	185	432
Global	40646.7	56659	3137.6	77.2	0.91	711	410

Figure 7: Lake distribution over glacier altitudinal range for each glacier region. Number of lakes (blue) and lake volume (orange) in 30 elevation bins, with the y-axis normalized to the largest bin. The normalized glacier altitudinal range represents a scale from 0 (lowest) to 1 (highest point of a glacier). Each lakes location on that scale is computed by normalizing the mean glacier surface elevation under which the lake is located. Blue and orange numbers in each subplot show the mean normalized altitude of the lake counts and volumes, respectively.

3.2.2 Part II

The 'realistic' nature of the topography is a key point being made several times in the manuscript. I agree that it appears 'realistic' in the examples I can see but I think it would be helpful to go into more detail/depth about what makes it realistic (and where it is not realistic – provide some detail about where and why such issues occur). On this point of understanding the realistic nature of the topography, it would also be useful to include more figures of where key types of features are modelled as being present – e.g. to back up the point on the examples of these features being present. This is particularly important given that many of them would then be home to the lakes subsequently.

RESPONSE: To address the request for a more detailed description of what makes TOPO-DE realistic, we refer to our reply to reviewer 1 who asked for a better justification of why we claim that TOPO-DE is superior to existing products. In brief, we show that TOPO-DE has the lowest MAE of all thickness products against GlaThiDa observations; that TOPO-DE is free of thickness-dependent biases seen in the products by [1] and [2]; that their unphysical artifacts can be identified on a large scale, in contrast to TOPO-DE which does not exhibit such artifacts; and that TOPO-DE is suitable for geological mapping of currently glacierized terrain, in contrast to existing products. All this information was added to the main text (see above for text changes).

In addition, to address the point of issues in TOPO-DE, a paragraph on limitations was added in the main text together with more detailed descriptions in the supplement. The added paragraph in the main text reads:

”Nevertheless, TOPO-DE inevitably contains inaccuracies arising from input data errors, physical shortcomings, and solution equifinality inherent to underconstrained ice thickness inversions (Supplementary Discussion). Biases in surface elevation change, mass balance, glacier outlines, velocities, and DEMs can locally affect inferred thickness and bed geometry [11, 12]. Meanwhile, the use of higher-order ice-flow physics substantially reduces physical shortcomings compared to SIA-based approaches [1, 2]. Errors are expected to be elevated for surging glaciers [13], in regions with few thickness and mass balance observations [3, 14], and for bed features smaller than approximately one ice thickness, which are fundamentally unconstrainable [15–17]. More ice thickness observations are critical to improve glacier bed inversions further, specifically in data-sparse regions such as High Mountain Asia, the Russian Arctic, and Subantarctic & Antarctic Islands. Our input-preparation and data-assimilation framework effectively minimizes and balances error sources, but residual artifacts naturally remain and can propagate into modeled lake locations and volumes. At large scales, we expect errors to average out, whereas localized applications should evaluate bed shape, ice thickness and lakes in light of these uncertainties. Importantly, the data assimilation framework used here makes TOPO-DE particularly well-suited to be refined by future improvements in input data sets.”

The text in the supplement reads:

”Biases in the dh/dt and mass balance products, most likely to be found in regions with few mass balance observations [14], lead to volume biases of individual glaciers. Errors in the conversion from elevation to mass change act in the same manner. However, the non-linear ice flow physics render the influence of such mass flux errors on modeled thicknesses and volumes small [12]. Erroneous glacier outlines that encompass ice-free terrain introduce a geometrical inconsistency between modeling domain and actual ice cover, typically resulting in overestimated ice thickness. Ice-free terrain as seen in a DEM may also show up inside glacier outlines if a DEM has a later acquisition date than the outlines, due to glacier retreat. On glacier complexes, incorrect ice divides compromise the mass budget closure of individual outlet glaciers because the delineated catchments do not correspond to the actual flow directions. This leads to excess mass in one outlet and mass deficits in another, with corresponding thickness biases. Mass shortages result in zero ice thicknesses near the glacier front - such areas are filled using a perfect-plasticity approach (see above) or by interpolation with associated elevated uncertainties. In fast-flowing areas of marine-terminating glaciers where we invert for basal friction ice velocity errors lead to too thick and too thin ice for negative and positive biases, respectively [12]. Bed shape errors are introduced where observed surface elevations correspond to features not produced by ice dynamics but by surface processes (e.g. large snow drifts, medial moraines, and debris). Our input preparation workflow (Supplementary Methods) is designed to minimize the influence of input data errors as much as possible though residual artifacts naturally remain. Thanks to higher-order ice flow physics, TOPO-DE suffers much less from shortcomings due to unmodeled processes than previous SIA-based studies that ignored horizontal stress gradients [1, 2]. However, we expect elevated errors for surging glaciers for which a comprehensive flow law is missing [13]. Furthermore, different time stamps of input datasets are problematic for surging glaciers because fast dynamical changes require highly time-synchronized inputs for accurately obtaining bed elevations [18]. The unavailability of a global inventory of lake-terminating glaciers forces us to model such glaciers as land-terminating, contributing to increased errors at and near their fronts. Finally, ice thickness inversions are generally underconstrained [15], implying that several solutions may explain the observa-

tions. Bed features smaller than at least one ice thickness cannot be resolved, setting a physical limit to the bed detail of any inversion [16, 17]. The modeled lakes are potentially influenced by all these challenges, introducing unavoidable uncertainty on their actual location and volume. ”

Finally, on the point of showing more figures of where key types of features are modeled as being present, we would like to point out that we already included three figures showing how bed shapes look in TOPO-DE in the manuscript (Fig. 1 and Extended Data Figs. 2 and 3 in the original manuscript). We find that adding more local examples would not say much, neither about the global distribution of certain landforms nor the quality of the dataset. In general, we find that a global analysis of landforms - which later reviewer comments also hint at - is outside the scope of this study. A comprehensive global-scale identification of U-shaped valleys, cirques, etc., requires careful geomorphological analysis of the landscape, probably complemented by some form of image recognition tool to handle the vast amounts of data. Such an undertaking is not feasible here. In contrast, potential future lakes are relatively easy to identify, allowing us for the first time to shed light on their global distribution. We do agree that a lot of potential lies in TOPO-DE to scan for more complex landforms, too, but this requires a dedicated study on its own. We have added this as an important goal for future research in the outlook sections (as part of a larger rewrite of the final paragraph, see below):

”TOPO-DE offers substantial opportunities for further characterization of the subglacial landscape, which warrant dedicated investigation in future studies.”

3.2.3 Part III

Associated with the realistic nature of the landscape, there is a good quantification to uncertainty outlined in the extended data, but in the main text of the manuscript the uncertainties are not really brought in where quantities are being mentioned. E.g. where ice volumes are outlined, what happens to those when uncertainty is accounted for? Its in the table but not really brought in in the text. How does that uncertainty impact the comparison of volumes against the work from other papers? It would be useful to have the associated assessment of uncertainty/accuracy included in the database. Moreover, more uncertainty occurs under particular data/modelling conditions, and so where such conditions occur you might have a grid that highlights you have less confidence in that area.

RESPONSE: We are not entirely sure what the reviewer refers to here, as volume uncertainties were given both in the text and in Fig. 4 (original manuscript) when we compare our ice volumes to Farinotti and Millan. Nevertheless, in the revised manuscript we have expanded the discussion on uncertainties when comparing our ice volumes to the previous work. Text changes include:

”The considerable disagreement with [2], where uncertainties barely overlap, may be a result of [2] predominantly relying on BedMachine [19] in this region, which likely produces too thin ice in poorly observed valley glaciers, as previously shown for the Antarctic peninsula [20]. In contrast, for the Greenland Periphery, our volume ($12.46 \pm 0.94 \times 10^3 \text{ km}^3$) is nearly identical to that of [2] ($12.54 \pm 3.95 \times 10^3 \text{ km}^3$), but $\sim 20\%$ lower than the $15.69 \pm 4.07 \times 10^3 \text{ km}^3$ estimated by [1], with their central estimate outside our uncertainty range. [1] is biased by about 90 m to thickness observations in this region, whereas our results are almost bias-free (4 m, Fig. S1). These examples from two regions highlight how our results provide much-needed clarification where the two previous estimates diverged.”

To evaluate point-thickness uncertainty for each RGI region we refer to the MEs and MAEs given in Extended Data Fig. 1 (now Fig. S1). To provide further transparency on uncertainties in TOPO-DE we have expanded the description of the validation experiment where we removed 50% of observations for calibration (see reply to reviewer 1 and associated text changes). In addition, we have added a more extensive discussion on limitations and likely error sources in the main text and in the supplement (see above for text changes). To produce a distributed map of uncertainty for our thicknesses is not possible because formal error propagation through the complex ice-dynamical model is not feasible and computational constraints limit our ability to conduct a representative ensemble spanning the full space of input data uncertainties.

3.3 Suggested improvements (minor)

Title: The landscapes are not currently ‘lake-rich’. They only would be if/when the glaciers disappear. Thus the title, which implies the lakes exist under the earths glacier is not quite appropriate. One option would be to use a more topographic term if you wish to talk about what is under the glaciers now (e.g. overdeepenings?) or alter the title to reflect the fact that the landscapes would hold lakes only once the glaciers have gone.

RESPONSE: We changed the title to: "Global glacier-free topography reveals large potential for future lakes in presently ice-covered terrain"

11: *'Retreating glaciers profoundly reshape landscapes. This could be confused with the idea of erosion under glaciers which also reshape landscapes. Here I assume that you are trying to say that the landscape looks a lot different once the glaciers disappear? If so, perhaps rephrase this sentence.*

RESPONSE: We rephrased to: "Glacier retreat transforms landscapes in polar and mountainous regions."

18: *The lake density calculation mentioned, which is conducted later in the paper, needs expanded upon at some stage in the paper. In particular, how does this compare to previously glaciated regions that are now ice free? E.g. is the number of lakes in your future deglaciated terrain similar to that of the most recently deglaciated terrain? At the moment, I feel like you compare it to all ice free terrain, whether previously glaciated or not. This is fine, but I also think you need the comparison against terrain that has been glaciated as this would be an interesting sense-check on the realistic nature of your dataset.*

RESPONSE: We conducted a literature search to find observations of lake density in formerly glaciated terrain, and found data for the Swiss and Austrian Alps [21, 22]. To provide another number, we calculated the glacier retreat over land on Svalbard using glacier outlines from 1936-1938 by [23] and the RGI, and compared that to lake formation estimates by [24] from those same years to 2008-2012 (noting that [24] only mapped ice-contact lakes). We added this paragraph:

"In the Alps, [21, 22] observed lake covers of 0.5% (Austria) and 0.9% (Switzerland) on land that became ice-free since the Little Ice Age. We estimate a potential future lake coverage of 0.9% in Central Europe, closely aligned with these historic values. For comparison, [25] modeled a potential future lake area of 45.2 km² in the Swiss Alps corresponding to a coverage of 4.7%, about four times higher than the historical reference. In Svalbard, [24] showed an increase in ice-marginal lakes of 72 km² between the 1930s and ~2010 while 2,144 km² land became ice-free [23] (3.3% lake coverage). Here, our estimate is 5.6% (including not only ice-marginal, but also pro-glacial lakes)."

15–18 vs. 19–20: *If the paper is restructured to put the ice volumes etc. first, then adjust the order of these groups of sentences.*

RESPONSE: As discussed above, we prefer to keep the original structure.

35: *'new'. Its not really 'new' landscape - its already there - just buried. So delete 'new'?*

RESPONSE: Done.

36–37: *grammatically awkward. The acceleration point could be added to end of previous sentence. Then the poorly known point could be added to the start of the next sentence where you say its poorly known and why that is important.*

RESPONSE: We rephrased to: "Indeed, land has already started to emerge from the retreating ice unveiling >2,000 km of previously ice-covered coastline in the Northern Hemisphere alone in the last 20 years [26] - a process that is expected to accelerate [27]. However, the topography of the emerging landscape is poorly known, which poses a severe limitation to our ability to sustainably manage these lands, ..."

37: *'severe': is it severe? Need to back this up with an indication of why its severe. E.g. what are the impacts of not being able to manage these things? Thy is it a limitation? If the lakes are to be the key point of this paper, then much more needs to be made of the importance of knowing where they may be in the future, and also what their importance may be. In addition – later – much more would need to be done in exploring where they are regionally and within regions and within what types of topography.*

RESPONSE: We expanded the lake analysis (see above). Meanwhile, at this point in the text, we find that it is sufficient to mention that poorly knowing the bed topography negatively affects planning of future hydropower in currently glaciarized terrain, projections for future species colonization, and hazards originating from the subglacier given that this is rather evident.

39: *'to mitigate potential hazards'. Such as what?*

RESPONSE: We have added: "...or to mitigate potential hazards, such as Glacier Lake Outburst Floods (GLOFs) from moraine- or bedrock-dammed lakes"

40–43: *This feels structurally awkward. Why is ice behaviour/volume not the first point being made? I say this*

because the exposed topography would come after the ice disappears and so logically would feel like that should be the way it is written.

RESPONSE: Given that we mention the observed emergence of lands first in this paragraph (which in turn connects to the previous paragraph), we find it fitting to discuss our insufficient knowledge of the future ice-free lands directly after. In our view, this is a research gap on its own, independent of the question of when exactly the lands will become ice-free or what the global glacier volume is.

47–48: *this needs expanded on slightly - are the discrepancies and uncertainties different between the Farinotti paper and the Millan paper? If so, say so directly. If not, indicate how discrepancies are known etc.*

RESPONSE: We refer to discrepancies between the Farinotti and Millan results, so one cannot say that the discrepancies are different between the two. To clarify, we have added: "... between the two studies..".

54: *perhaps slight mention of what is unrealistic about them? How is it known they are unrealistic?*

RESPONSE: We note that this is discussed later (1.93 - 106 in the original manuscript) which we indicated by adding "(see below)".

87: *great. These are expected, but sometimes they are expected in particular places within a glacier catchment. Therefore, it may be worth mentioning where you tend to find such features. E.g. are the cirques always found at the heads of the valleys? Are the overdeepenings found in most trunks or only where confluence of glacier flow occurs? Basically, can you describe more about where such features are found within the landscapes? Do they also tend to be found in particular quantities in particular regions? Expand a bit if possible.*

RESPONSE: As mentioned above we substantially expanded the analysis of the lakes including a description of where we find them in the landscape, and how their distribution differs between regions. For other landforms, we would like to refer to our previous reply and the difficulties of identifying and mapping these features on a global scale. As discussed above, we have added future work on further characterizing the subglacial landscape as an important research aim in the outlook section (see above for text changes).

93: *Is isostatic unloading from ice removal accounted for in TOPO-DE? I realise it may not make a huge difference to local features such as lakes but in some areas it may influence things like the shoreline or the overall regional gradients in a landscape. Accounting for this would be important to maximise the usefulness of TOPO-DE. Alternatively, a statement indicating it is not accounted for would be appropriate to add in the methods section.*

RESPONSE: Isostatic unloading is not accounted for as this is a complex process which 1) depends on poorly constrained properties of Earth's inner structure, and 2) is dependent on the rate of deglaciation and as such on a process which we explicitly do not model here. As suggested, we have added this information to the Methods: "Glacial isostatic adjustment as a time-dependent process unfolding during and after deglaciation is not accounted for - the reconstructed beds reflect the present-day shape of the subglacial topography."

107–108: *I think if inaccuracies are going to be mentioned then it would be more open to then provide a bit of clearer information on that - perhaps in the supplementary information section. And with some example figures to go with text describing the issues.*

RESPONSE: We agree with the reviewer and have expanded this paragraph as well as added a section on limitations in the supplement (see above for text changes).

110–111: *It would seem like this could be a good idea for a supplementary figure here to show a figure of topography in an already well measured area vs. your new inverted topography (in a version you create without control data from that particular location). A figure would be particularly useful given you use some supplementary figures to critique (rightly-so) the previously existing topographic reconstructions - so therefore an appropriate exploration/critique of your own dataset should be presented as some text with potentially a supplementary figure.*

RESPONSE: We have added Fig. 8 to the supplement which shows the ice thickness of Aletsch glacier modeled without using observations from that glacier, with overlain known ice thicknesses. A sentence in the methods section was added stating: "Fig. S7 shows the example of Aletsch glacier and surrounding glaciers in the Swiss Alps modeled without using their thickness observations." In addition, we added a section on limitations to provide a critique of our own dataset (see above for text changes).

128–130: *perhaps link this back to specific types of features in that landscape? E.g. overdeepenings are larger under the large systems vs. cirques being more dominant in the steeper smaller systems?*

Figure 8: Modeled and observed ice thickness of Aletsch glacier and surrounding glaciers from the validation experiment. All thickness observations on those glaciers (underlain by white lines) were not used to obtain these results.

RESPONSE: Again, please see our expanded analysis of the lakes which includes a discussion on where we tend to find large overdeepenings vs. smaller lakes in glacial cirques.

132–134: what is average lake size and total lake counts in such areas? And average lake volumes? Suggest quantifying in the text.

RESPONSE: We have added such regionally aggregated numbers in Table 1 and discussed them in the expanded lake analysis (see above).

142–143: If you are going to talk about GLOF then one key point is that many recent GLOF events have been in moraine-dammed proglacial lakes. Presumably, however, you are making the assumption that the deglaciated bed is bedrock and thus that the lakes are impounded not by moraine but by rock. As a result, I wonder if you are over-emphasising the potential GLOF risk in a deglaciated environment? Perhaps the mode of impounding could be discussed? Discuss the GLOF risk in more detail overall if lakes are a key point of the paper.

RESPONSE: See above, we reconstruct the bed topography that gave rise to the observed glacier dynamics. Consequently, we did not make any assumption on whether a glacier is underlain by bedrock or sediment. Indeed, we do see indications that many lakes are going to form behind moraines, highlighting the importance of our study for future GLOF risk. This information as well as a discussion of different modes of impounding was added in the expanded lake analysis (see revised manuscript for text changes).

145: ‘Site specific opportunities and constraints must be assessed carefully’. This feels rather vague. Can you be more specific?

RESPONSE: We have rephrased to: ”For the sustainable development of each site, case-specific opportunities and constraints must be carefully assessed, taking into account various factors including technological feasibility, environmental vulnerability, and social acceptance [28, 29]”

153: Perhaps worth indicating why (e.g. they did not recover the bed in appropriate detail or their assumptions were too limiting).

RESPONSE: Most of the previous studies were based on Volume-Area scaling which is a highly simplified description of ice dynamics. In addition, early inversions were limited by a lack of available input data. We have added: ”... due to considerably simpler methodological approaches and a lack of high-quality input data”.

155–158: While reflecting on differences, it would be useful to indicate why such differences occur. For example, is it because you predict overdeepenings better, or thinner/thicker ice in valleys compared to previous estimates etc? In other words, where in the landscape is the extra ice found in comparison to reference 13? Just thicker overall? Or more ice in valleys due to overdeepened topography? Explore this a bit.

RESPONSE: Unfortunately, Millan et al. (2022) (reference 13) do not provide distributed thickness maps for most of RGI region 19 (Antarctic and Subantarctic), complicating such an analysis. The reason for that is presumably that they have relabeled many glaciers as belonging to the ice sheet, and relied on BedMachine to calculate glacier volumes (see the author correction to their article and [30]). However, in the literature, we find indications that their product underestimates ice thicknesses in the Antarctic and Subantarctic, which we added to the text: ”The considerable disagreement with [2], where uncertainties barely overlap, may be a result of [2] predominantly relying on BedMachine [19] in this region, which likely produces too thin ice in poorly observed valley glaciers, as previously shown for the Antarctic peninsula [20].”

173–174: This is a tagline that is used in the paper a couple of times, but its not really explored much apart from the GLOF component. Is more exploration possible?

RESPONSE: We would like to point out that assessments of hydropower potential and ecological colonization are highly complex, with the two studies attempting this task (ref. 9 and 10 in original manuscript) both having appeared in Nature. Similarly, land management and safety in glacier tourism need to be studied in detail on local scales to yield meaningful results. As such, we deem a detailed investigation of these aspects to be outside the scope of this study, although we are convinced that our product will form a valuable input dataset to such work.

175–181: This is where you could do more exploration of the above. But in any case, these sentences feel rather vague in terms of what the new database is really going to be useful for. I think a better case for the impacts of this work could be made. You do pick up on the key point to future global modelling, but in an overly-technical manner. This paragraph needs to be made more accessible for a wide readership as is the case in Nature Communications.

RESPONSE: We reworded the section as follows:

"The new subglacial topographies provide critical data for a wide range of disciplines including glaciology, hydrology, ecology, geology, and geomorphology. As glaciers retreat in a warming climate, improved knowledge of emerging landscapes and lakes is essential for land management, hydropower planning, safety in glacier tourism, as well as hazard and ecosystem change assessments. TOPO-DE offers substantial opportunities for further characterization of the subglacial landscape, which warrant dedicated investigation in future studies. Our large-scale analysis highlights major GLOF hazards associated with potential future lakes in steep mountain terrain, particularly large moraine-dammed lakes in High Mountain Asia. More detailed, localized investigations will be essential to further refine this picture and to generate actionable data for mitigating such mountain hazards under climate change. Future glacier simulations based on the new bed product will be instrumental in constraining the timing of glacier retreat and the associated emergence of proglacial lakes. Together with the revised ice-volume estimates, which diverge substantially from previous studies across many glacier regions, future glacier projections based on TOPO-DE are expected to reveal new regional patterns of deglaciation with associated improved estimates of sea level rise.

More broadly, our results underscore the value of computationally efficient, higher-order ice-flow models in inversion frameworks that exploit the growing availability of high-resolution remote sensing data, paving the way for global glacier modeling beyond the era of shallow ice flow physics and one-dimensional flow line models."

208: is a particular type of friction/sliding therefore assumed?

RESPONSE: IGM includes a Weertman-type friction law. We added this information as follows: "... we additionally invert for a basal friction coefficient c of a Weertman-type friction law implemented in IGM[31]."

212: There is uncertainty in ITS_LIVE. How is that translated through to the inversion?

RESPONSE: We purposefully use velocity observations only in fast-flowing ($> 100 \text{ m yr}^{-1}$) areas of marine-terminating glaciers where the data typically is most reliable. We thus avoid common issues on small, slow glaciers where velocity observations often are noisy, or in low-contrast accumulation areas where feature tracking with optical satellites is challenging. Beyond that, in our added discussion on uncertainties and limitations in the supplement we mention: "In fast-flowing areas of marine-terminating glaciers where we invert for basal friction ice velocity errors lead to too thick and too thin ice for negative and positive biases, respectively [12]."

224–225: this feels vague. What is meant by neighbouring regions?

RESPONSE: This was previously mentioned in the Supplementary Material (1.92-98 original manuscript). We now added this information here as well: "For regions lacking point bed observations parameter values are assigned based on neighboring regions. These regions are Iceland where the mean of the parameters from Greenland and Svalbard are used, the Russian Arctic where the Svalbard parameters are taken, and South Asia West where the values from Central Asia are used."

243: Would that tend towards undershoots or overshoots? Does it have a tendency to put particular shapes into the topography where such interpolation is used?

RESPONSE: This depends on the specific setting of each glacier. It does contribute to larger uncertainties in such areas which we mention in the added discussion on uncertainties and limitations (see above for text changes).

277–280: A map-based figure to show how the new database compares with the measured thicknesses in one of these areas could be a useful one. It would allow better understanding of how well the approach does in estimating the thickness and bed.

RESPONSE: We added Fig. 8 showing Aletsch glacier whose observations were not used during the experiment where 50% of thickness observations were withheld, alongside a sentence on that in the Methods (see above for text changes).

Extended Data Table 2: This table shows how much RGI appears to differently estimate SLE compared to your work. In the text at present, you give the example of the subantarctic which is pretty much the only one where RGI and your estimate are close. Therefore I suggest an additional sentence in the text which clearly indicates that RGI has a strong tendency to significantly overestimate or underestimate compared to your work.

RESPONSE: We believe this comments refers to Farinotti, not the RGI. It is true that in the majority of regions, our volumes are closer to Millan than Farinotti. Aggregated globally, our volume is almost exactly in the middle between the two studies. We added in the text: "Overall, our volumes agree more closely with [2] than with [1] in the majority of regions."

On that point, you quote the mean differences in parentheses here between your work and others. The means are not provided for RGI - it would be interesting to have that. But even then, the means are not necessarily helpful because there are many + or - values that are quite large depending on region. Would it be useful to separate our mean underestimations and mean overestimations? And the same for Millan. Just a thought - might pull our whether the underestimates or the overestimates are the key differences in the datasets.

RESPONSE: We are not entirely sure how to understand this comment. What mean of the RGI is meant? Or is Farinotti meant as in the previous comments? What is meant by "our mean underestimations and overestimations"? We added the mean thicknesses from this study as a column in Supplementary Table 1).

References: These seem to cover the important bases.

References

1. Farinotti, D. *et al.* A consensus estimate for the ice thickness distribution of all glaciers on Earth. *Nature Geoscience* **12**, 168–173. doi:10.1038/s41561-019-0300-3 (Mar. 2019).
2. Millan, R., Mouginito, J., Rabatel, A. & Morlighem, M. Ice velocity and thickness of the world's glaciers. *Nature Geoscience* **15**, 124–129. doi:10.1038/s41561-021-00885-z (Feb. 2022).
3. GlaThiDa Consortium. *Glacier Thickness Database 3.1.0* Zürich, Switzerland, 2020.
4. Welty, E. *et al.* Worldwide version-controlled database of glacier thickness observations. *Earth System Science Data* **12**, 3039–3055. doi:10.5194/essd-12-3039-2020 (Nov. 2020).
5. Fürst, J. J. *et al.* Application of a two-step approach for mapping ice thickness to various glacier types on Svalbard. *The Cryosphere* **11**, 2003–2032. doi:10.5194/tc-11-2003-2017 (Sept. 2017).
6. Harrison, J. C. & Jackson, M. P. *Bedrock geology, Strand Fiord-Expedition Fiord area, western Axel Heiberg Island, northern Nunavut (parts of NTS 59E, F, G, and H)* tech. rep. 5590 (Geological Survey of Canada, 2008).
7. Farinotti, D., Round, V., Huss, M., Compagno, L. & Zekollari, H. Large hydropower and water-storage potential in future glacier-free basins. *Nature* **575**, 341–344. doi:10.1038/s41586-019-1740-z (Nov. 2019).
8. Blatter, H. Velocity and stress fields in grounded glaciers: a simple algorithm for including deviatoric stress gradients. *Journal of Glaciology* **41**, 333–344. doi:10.3189/S002214300001621X (1995).
9. Pfeffer, W. T. *et al.* The Randolph Glacier Inventory: a globally complete inventory of glaciers. *Journal of Glaciology* **60**, 537–552. doi:10.3189/2014JoG13J176 (2014).
10. RGI Consortium. *Randolph Glacier Inventory – A Dataset of Global Glacier Outlines: Version 6.0* (NSIDC: National Snow and Ice Data Center, Boulder, Colorado USA, 2017).
11. Farinotti, D. *et al.* How accurate are estimates of glacier ice thickness? Results from ITMIX, the Ice Thickness Models Intercomparison eXperiment. *The Cryosphere* **11**, 949–970. doi:10.5194/tc-11-949-2017 (Apr. 2017).
12. Frank, T., van Pelt, W. J. J. & Kohler, J. Reconciling ice dynamics and bed topography with a versatile and fast ice thickness inversion. *The Cryosphere* **17**, 4021–4045. doi:10.5194/tc-17-4021-2023 (Sept. 2023).
13. Thøgersen, K., Gilbert, A., Schuler, T. V. & Malthe-Sørensen, A. Rate-and-state friction explains glacier surge propagation. *Nature Communications* **10**, 2823. doi:10.1038/s41467-019-10506-4 (June 2019).
14. WGMS. *Fluctuations of Glaciers Database* World Glacier Monitoring Service, Zürich, Switzerland, 2022. doi:10.5904/wgms-fog-2022-09.
15. Bahr, D. B., Pfeffer, W. T. & Kaser, G. Glacier volume estimation as an ill-posed inversion. *Journal of Glaciology* **60**, 922–934. doi:10.3189/2014JoG14J062 (2014).
16. Raymond, M. J. & Gudmundsson, G. H. On the relationship between surface and basal properties on glaciers, ice sheets, and ice streams. *Journal of Geophysical Research: Solid Earth* **110**. doi:10.1029/2005JB003681 (2005).
17. Gudmundsson, G. H. & Raymond, M. On the limit to resolution and information on basal properties obtainable from surface data on ice streams. *The Cryosphere* **2**, 167–178. doi:10.5194/tc-2-167-2008 (Dec. 2008).

18. Van Pelt, W. & Frank, T. New glacier thickness and bed topography maps for Svalbard. *The Cryosphere* **19**, 1–17. doi:10.5194/tc-19-1-2025 (Jan. 2025).
19. Morlighem, M. *et al.* Deep glacial troughs and stabilizing ridges unveiled beneath the margins of the Antarctic ice sheet. *Nature Geoscience* **13**, 132–137. doi:10.1038/s41561-019-0510-8 (Feb. 2020).
20. Shahateet, K. *et al.* A reconstruction of the ice thickness of the Antarctic Peninsula Ice Sheet north of 70° S. *The Cryosphere* **19**, 1577–1597. doi:10.5194/tc-19-1577-2025 (Apr. 2025).
21. Mölg, N. *et al.* Inventory and evolution of glacial lakes since the Little Ice Age: Lessons from the case of Switzerland. *Earth Surface Processes and Landforms* **46**, 2551–2564. doi:10.1002/esp.5193 (2021).
22. Buckel, J., Otto, J. C., Prasicek, G. & Keuschnig, M. Glacial lakes in Austria - Distribution and formation since the Little Ice Age. *Global and Planetary Change* **164**, 39–51. doi:10.1016/j.gloplacha.2018.03.003 (May 2018).
23. Geyman, E. C., J. J. van Pelt, W., Maloof, A. C., Aas, H. F. & Kohler, J. Historical glacier change on Svalbard predicts doubling of mass loss by 2100. *Nature* **601**, 374–379. doi:10.1038/s41586-021-04314-4 (Jan. 2022).
24. Wieczorek, I., Strzelecki, M. C., Stachnik, L., Yde, J. C. & Malecki, J. Post-Little Ice Age glacial lake evolution in Svalbard: inventory of lake changes and lake types. *Journal of Glaciology* **69**, 1449–1465. doi:10.1017/jog.2023.34 (Oct. 2023).
25. Steffen, T., Huss, M., Estermann, R., Hodel, E. & Farinotti, D. Volume, evolution, and sedimentation of future glacier lakes in Switzerland over the 21st century. *Earth Surface Dynamics* **10**, 723–741. doi:10.5194/esurf-10-723-2022 (July 2022).
26. Kavan, J. *et al.* New coasts emerging from the retreat of Northern Hemisphere marine-terminating glaciers in the twenty-first century. *Nature Climate Change*, 1–10. doi:10.1038/s41558-025-02282-5 (Mar. 2025).
27. Rounce, D. R. *et al.* Global glacier change in the 21st century: Every increase in temperature matters. *Science* **379**, 78–83. doi:10.1126/science.abo1324 (Jan. 2023).
28. Haeberli, W. *et al.* New lakes in deglaciating high-mountain regions – opportunities and risks. *Climatic Change* **139**, 201–214. doi:10.1007/s10584-016-1771-5 (Nov. 2016).
29. Kellner, E. Social Acceptance of a Multi-Purpose Reservoir in a Recently Deglaciating Landscape in the Swiss Alps. *Sustainability* **11**, 3819. doi:10.3390/su11143819 (Jan. 2019).
30. Hock, R., Maussion, F., Marzeion, B. & Nowicki, S. What is the global glacier ice volume outside the ice sheets? *Journal of Glaciology* **69**, 204–210. doi:10.1017/jog.2023.1 (Feb. 2023).
31. Jouvét, G. & Cordonnier, G. Ice-flow model emulator based on physics-informed deep learning. *Journal of Glaciology*, 1–15. doi:10.1017/jog.2023.73 (Sept. 2023).